# miR-329– and miR-495–mediated Prr7 down-regulation is required for homeostatic synaptic depression in rat hippocampal neurons

Michiko O Inouye, David Colameo, Irina Ammann, Jochen Winterer, Gerhard Schratt

Homeostatic synaptic depression (HSD) in excitatory neurons is a cell-autonomous mechanism which protects excitatory neurons from over-excitation as a consequence of chronic increases in network activity. In this process, excitatory synapses are weakened and eventually eliminated, as evidenced by a reduction in synaptic AMPA receptor expression and dendritic spine loss. Originally considered a global, cell-wide mechanism, local forms of regulation, such as the local control of mRNA translation in dendrites, are being increasingly recognized in HSD. Yet, identification of excitatory proteins whose local regulation is required for HSD is still limited. Here, we show that proline-rich protein 7/transmembrane adapter protein 3 (Prr7) down-regulation in dendrites of rat hippocampal neurons is necessary for HSD induced by chronic increase in network activity resulting from a blockade of inhibitory synaptic transmission by picrotoxin (PTX). We further identify two activity-regulated miRNAs, miR-329-3p and miR-495-3p, which inhibit Prr7 mRNA translation and are required for HSD. Moreover, we found that Prr7 knockdown reduces expression of the synaptic scaffolding protein SPAR, which is rescued by pharmacological inhibition of CDK5, indicating a role of Prr7 protein in the maintenance of excitatory synapses via protection of SPAR from degradation. Together, our findings highlight a novel HSD mechanism in which chronic activity leads to miR-329– and miR-495–mediated Prr7 reduction upstream of the CDK5-SPAR pathway.

## Introduction

Homeostatic synaptic depression (HSD) is a type of homeostatic plasticity by which excitatory neurons compensate for increased network activity to maintain a physiological range of excitatory transmission (reviewed in Turrigiano [2008], Yu and Goda [2009], and Turrigiano [2012]). Adaptive mechanisms to maintain neuronal homeostasis include changes in synaptic AMPA receptors (Seeburg et al, 2008) and spine number (Kirov et al, 1999; Wierenga et al, 2006). Abnormal dendritic spine density and altered AMPAR internalization have been suggested in epilepsy (Isokawa et al, 1997), schizophrenia (Glantz & Lewis, 2000), and autism spectrum disorder (Hutsler & Zhang, 2010), as well as in disease models of fragile X (Fmr1-knockout) (Jawaid et al, 2018) and Rett syndrome (Mecp2-mutant) (Chao et al, 2007), highlighting the importance of HSD regulation for neuronal homeostasis.

Although most of the studies on HSD have used GABA-receptor antagonists (e.g., picrotoxin [PTX] and bicuculline) to investigate HSD in response to network-wide stimulation, a study employing optogenetic methods of stimulation demonstrated that the mechanism of HSD is cell-autonomous (Goold & Nicoll, 2010). Namely, upon 24-h photostimulation of channel rhodopsin-2 (ChR2)–expressing CA1 pyramidal neurons, lowered mEPSC frequency and dendritic spine number were observed, indicating that individual neurons possess intrinsic mechanisms to regulate their synapse number in response to chronic activity. The spine loss in HSD is supported by a number of other studies using either optogenetics (Mendez et al, 2018) or pharmacological stimulation (Pak & Sheng, 2003; Fiore et al, 2014; Chowdhury et al, 2018).

Pathways underlying HSD have been examined in detail. In a well-studied mechanism, elevated synaptic activity first causes L-type voltage-gated calcium channel opening and NMDAR activation, which results in calcium influx (Pak & Sheng, 2003; Goold & Nicoll, 2010). Calcium binds calmodulin, which initiates a cascade of CaM kinases (Wayman et al, 2008), of which CaMKK and CaMK4 are required for HSD (Goold & Nicoll, 2010). The cascade transcriptionally activates polo-like kinase 2 (Plk2/SNK), a member of the polo family of serine/threonine protein kinases, over a time scale of hours (Kauselmann et al, 1999; Pak & Sheng, 2003). The induced Plk2 is targeted to dendritic spines and binds to a PSD-95 interacting factor, spine-associated Rap guanosine triphosphatase (GTPase)–activating protein (GAP) (SPAR), which has been "primed" for Plk2 binding by CDK5, a proline-directed kinase (Seeburg et al, 2008). The Plk2-SPAR binding results in proteasome-directed SPAR degradation,

Department of Health Science and Technology, Laboratory of Systems Neuroscience, Institute for Neuroscience, Swiss Federal Institute of Technology ETH, Zürich, Switzerland

Correspondence: gerhard.schratt@hest.ethz.ch

which has downstream effects on actin dynamics and Rap signaling, eventually leading to AMPAR and NMDAR removal and loss of spines. Although the Plk2-SPAR association is linked to synaptic AMPAR reduction without any reported preference to GluA1 or GluA2 subunits, a separate kinase-independent pathway in which Plk2 binds to N-ethylmaleimide-sensitive fusion protein (NSF) which is selective to GluA2 subunit removal has been observed (Evers et al, 2010).

Although there is evidence that excitatory synapses scale in a cell-wide, uniform manner during HSD, theoretical consideration invoke the existence of additional local dendritic mechanisms to assure proper information processing (Rabinowitch & Segev, 2006a, 2006b). In fact, several local dendritic mechanisms which are engaged during homeostatic plasticity have been recently described. For example, chronic inactivity with NMDAR inhibition leads to retinoic acid signaling and stimulation of local GluA1 synthesis (Aoto et al, 2008; Poon & Chen, 2008). Similarly, homeostatic upscaling via tetrodotoxin (TTX) and AMPAR/NMDAR blockade has also been shown to stimulate local protein synthesis (Sutton et al, 2007), including GluA1 accumulation (Sutton et al, 2006). In the context of HSD, miR-134-mediated local translation of Pumilio-2 (Pum2) mRNA upstream of Plk2 activation was reported (Fiore et al, 2014). The involvement of other miRNAs, namely, miR-129 (Rajman et al, 2017) and miR-485 (Cohen et al, 2011) in HSD further suggest the importance of local translation in this process. In addition, dendrite-specific regulation of excitatory proteins in the context of HSD has been demonstrated on a multi-omics scale (Colameo et al, 2021). Indeed, such local translational mechanisms could explain the spatial specificity of HSD, as demonstrated by homeostatic regulation in individual dendritic compartments (Rabinowitch & Segev, 2006a). The spatial specificity further extends to the synapse level as scaling depends on the spatial patterns of synaptic potentiations (Rabinowitch & Segev, 2006b). Considering the mounting evidence for local regulation of proteins in plasticity mechanisms such as long-term potentiation (LTP) in both in vitro and in vivo contexts (Miller et al, 2002; Lyles et al, 2006), there is a need for identifying other proteins that are locally regulated in HSD.

In the present study, we investigate the expression regulation and function of proline-rich 7/transmembrane adapter protein 3 (Prr7) in the context of HSD induced by chronic activity. Prr7 is localized in neuronal dendrites of hippocampal neurons (Murata et al, 2005; Kravchick et al, 2016; Lee et al, 2018) and the postsynaptic density in rodent brains (Jordan et al, 2004; Yoshimura et al, 2004; Murata et al, 2005), suggesting that it could play an important role in HSD. Functionally, exosomally secreted Prr7 induces synapse elimination in hippocampal neurons (Lee et al, 2018), whereas the NMDA receptor mediated induction of excitotoxicity is accompanied by a translocation of Prr7 from the synapse to the nucleus, followed by a triggering of Jun-dependent apoptotic pathway (Kravchick et al, 2016). However, whether synaptically localized Prr7 is involved in activity-dependent forms of synaptic plasticity, for example, HSD, is unknown.

Here, we report that the down-regulation of Prr7 at both RNA and protein levels is required for dendritic spine elimination during HSD induced by chronic activity and that the dendritic reduction of Prr7 is regulated posttranscriptionally by miR-329 and miR-495. Furthermore, our results suggest that the miR-329/495/Prr7 interaction

ties in with the previously described SPAR/CDK5 pathway involved in homeostatic plasticity.

# Results

## Prr7 mRNA and protein are down-regulated locally in processes by chronic activity

Prr7 has previously been identified as a synaptic protein and implicated in the control of excitatory synapse formation, but its role in synaptic plasticity, for example, HSD, is unknown. Specifically, whether the subcellular expression of Prr7 changes during HSD has not been investigated. To determine if Prr7 expression is regulated in HSD, we first examined Prr7 mRNA expression levels in mature (DIV21) primary rat hippocampal cells treated with either mock (Ethanol) or the GABA-A receptor antagonist picrotoxin (PTX) for 48 h. This is a well-established experimental paradigm to induce HSD in vitro, as demonstrated by expected changes in spike frequency, EPSC amplitude, and expression of the GluA1 subunit of AMPARs (Ibata et al, 2008; Seeburg et al, 2008; Evers et al, 2010; Bateup et al, 2013; Fiore et al, 2014; Rajman et al, 2017). Using qRT-PCR, we observed a decrease in Prr7 mRNA levels in whole-cell extracts of PTX, compared with mock-treated hippocampal neurons (Fig 1A), which was similar in magnitude to GluA1 mRNA which was previously shown to be down-regulated during HSD.

Previous studies indicate that in addition to neuron-wide changes, local alterations in gene expression in the synapto-dendritic compartment might also be involved in homeostatic plasticity (Sutton et al, 2007; Colameo et al, 2021). To determine local expression changes, we used a compartmentalized culture system as previously described (Bicker et al, 2013), which allowed separate measurements of Prr7 RNA expression in mock versus PTX-treated cells in cell bodies and processes (which are mainly represented by dendrites), respectively. Therefore, we observed a significant decrease in Prr7 mRNA levels in the dendrite compartment upon PTX treatment (Fig 1B). Prr7 levels in the cell body compartment were more variable but also trended downward by PTX, consistent with our observations in whole-cell extracts.

We further probed for Prr7 protein expression by immunoblotting whole-cell (Fig 1C) and compartmentalized (Fig 1D) protein extracts from PTX and mock-treated hippocampal neurons. Although there was a general downward trend in Prr7 protein levels upon PTX treatment, the effect was most pronounced and statistically significant in dendrites (Fig 1D), providing further support for an important contribution of local regulatory mechanism engaged in the control of Prr7 expression during HSD.

To further corroborate the observed subcellular differences in Prr7 regulation in neurons, we additionally analyzed Prr7 protein levels through Prr7 immunostaining of GFP-transfected hippocampal neurons which were either mock- or PTX-treated (Fig 1E). Therefore, we used a commercial Prr7 antibody whose specificity was validated by the presence of reduced signal intensity in Prr7 knockdown cells (Fig S1A–C). We measured the average Prr7 puncta intensities within whole cell, cell body, and dendrite (whole cell with cell body removed) selections using GFP as a mask. Consistent

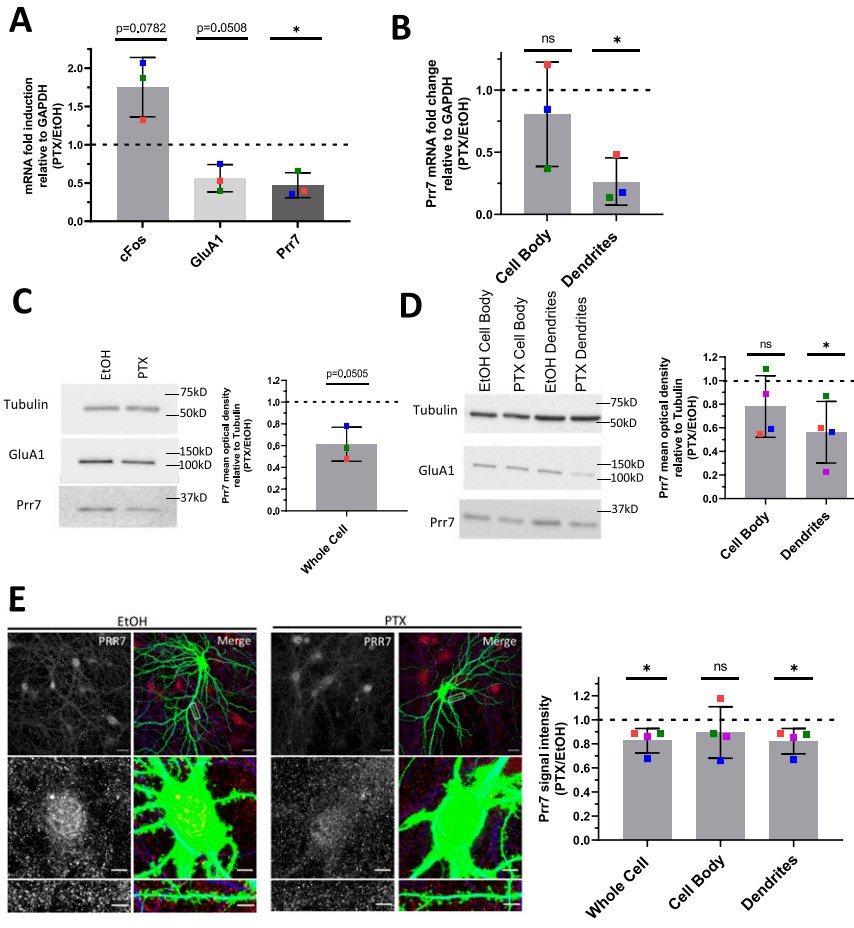

**Figure 1. Global and local Prr7 down-regulation at both RNA and protein levels by chronic activity.**
**(A)** Prr7 mRNA levels relative to GAPDH in hippocampal rat neurons treated with 100 μM PTX or EtOH (1:500 volume) at DIV17 for 48 h and lysed for RNA extraction on DIV19. *P = 0.0303. **(B)** Prr7 mRNA levels in compartmentalized hippocampal cell samples treated at DIV19 with 100 μM PTX or EtOH (1:500 volume) for 48 h (ns P = 0.5059, *P = 0.0216). **(C, D)** Prr7 protein levels as measured by mean optical band density relative to tubulin in (C) whole-cell hippocampal cell samples and (D) compartmentalized hippocampal rat cultures treated at DIV19 with 100 μM PTX or EtOH (1:500 volume) for 48 h (ns P = 0.1910, *P = 0.0441). **(E)** Representative whole-cell, cell body, and dendrite images showing Prr7 expression (gray scale, left panels) and merged Prr7 (red), GFP (green), and Map2 (blue) signals (right panels) in GFP-transfected hippocampal neurons treated with EtOH or PTX for 48 h. Scale bars = 20 μm (whole-cell images) and 5 μm (cell body and dendrite close-ups). On the right, average Prr7 punctum intensity in GFP-transfected (150 ng) cell body or dendrites selection of hippocampal rat neurons treated with PTX or EtOH on DIV19 for 48 h. Each point represents the grand average for the 7–9 cells imaged in a single experiment (*P = 0.0417 [whole cell], ns P = 0.3988 [cell body], *P = 0.0427 [dendrites]). For all bar graphs, data = mean normalized to EtOH condition ± SD, n = 3–4, *P < 0.05, one-sample t test with hypothetical mean set to 1. Colors of points represent data from the same independent experiment. Source data are available for this figure.

with the Western blot data, reduction in Prr7 puncta intensity upon PTX was most robustly observed in neuronal dendrites, whereas analysis of cell bodies only revealed a nonsignificant reduction of the Prr7 signal (Fig 1E, right panel). This decrease was homogenous along the dendrites because no difference in Prr7 down-regulation was detected between proximal versus distal dendrites (Fig S2A and B).

Notably, no significant differences in neither Prr7 mRNA nor protein levels between the cell body and dendritic compartments were observed in mock-treated neurons at baseline (Fig S3A–C), indicating that mechanisms leading to Prr7 down-regulation are specifically engaged during HSD.

### Down-regulation of Prr7 is necessary and sufficient for spine density reduction during HSD

Based on our finding that Prr7 is down-regulated during HSD, we asked whether Prr7 knockdown is sufficient to induce HSD. Prr7 knockdown was achieved using transfection of a Prr7 shRNA expressing plasmid based on a previously published Prr7 targeting sequence (Kravchick et al, 2016). We confirmed efficient and specific knockdown of Prr7 using the generated construct (Fig S4A and B). Prr7 knockdown did not adversely affect cell health based on unaltered cell morphology between control and Prr7 shRNA-transfected neurons (Fig S1B).

Using the validated shRNA construct, we found that Prr7 loss in hippocampal neurons led to a significant decrease in dendritic spine density, to levels comparable to those induced by PTX (Fig 2A). To determine whether spine density reduction was specific to Prr7 knockdown and not caused by off-target effects of the shRNA used, we generated an shRNA-resistant Prr7 expression construct (validations shown in Figs S5A–C and S6A and B). The spine density reduction from Prr7 shRNA was rescued when the shRNA-resistant Prr7 expression construct was introduced (Fig 2B). Moreover, a significant (~34%) increase in spine density was observed upon Prr7 overexpression alone. Furthermore, overexpression of Prr7 in hippocampal neurons led to complete prevention of spine density reduction in the presence of PTX (Fig 2C).

Because AMPAR degradation is a hallmark of HSD, we studied the effect of Prr7 knockdown on the protein levels of the GluA1 subunit of AMPA-type glutamate receptor (AMPAR). Knockdown of Prr7 in cortical neurons led to a reduction in total GluA1 protein levels as judged by immunoblotting (Fig 2D). In addition, in hippocampal neurons, GluA1 whole-cell puncta intensity and surface GluA1 puncta number were reduced in the Prr7 knockdown condition based on immunostaining (Fig 2E and F). Together, these observations suggest that loss of Prr7 might be sufficient to induce a reduction in spine density and GluA1, both of which are commonly observed during HSD.

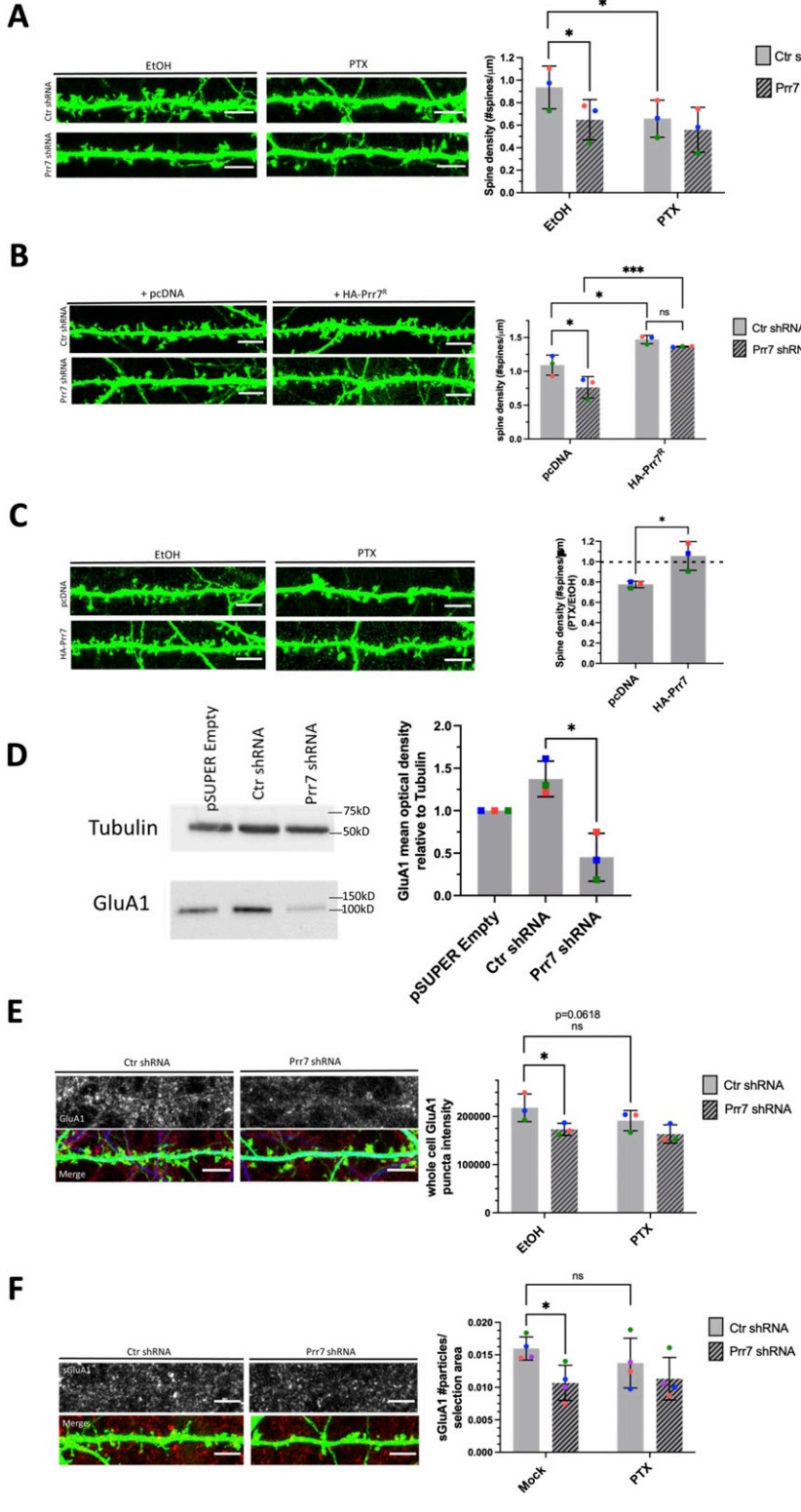

**Figure 2. Prr7 down-regulation is necessary and sufficient for homeostatic synaptic depression (HSD).**
**(A)** Spine densities of hippocampal rat neurons transfected with GFP (150 ng) and either control (plain bars) or Prr7 shRNA vector (7.5 ng pSUPER, patterned bars) on DIV13, treated at DIV19 with 100 $\mu$M PTX or EtOH (1:500 volume) for 48 h, with representative GFP images showing dendrites for each condition (Ctr EtOH versus Prr7 EtOH: $P$ = 0.0175; Ctr EtOH versus Ctr PTX: $P$ = 0.0187). **(B)** Spine densities of hippocampal neurons transfected with GFP (150 ng), either control (plain bars) or Prr7 shRNA (7.5 ng, patterned bars) and either pcDNA or shRNA-resistant Prr7 construct (400 ng) on DIV13 and fixed on DIV20-21 (Ctr pcDNA versus Prr7 pcDNA: $P$ = 0.0309; Ctr pcDNA versus Ctr HA-Prr7[R]: $P$ = 0.0146; Prr7 pcDNA versus Prr7 HA-Prr7[R]: $P$ = 0.0008; Ctr HA-Prr7[R] versus Prr7 HA-Prr7[R]: $P$ = 0.6899). For these shRNA experiments, data = mean ± SD, n = 3, *$P$ < 0.05, ***$P$ < 0.001, two-way ANOVA with Tukey's post hoc HSD test. **(C)** Spine densities of hippocampal neurons transfected with GFP (150 ng) and either pcDNA or HA-Prr7 construct (400 ng) on DIV13 and treated at DIV19 with EtOH or PTX for 48 h. Data = mean normalized to EtOH condition ± SD, n = 3, *$P$ = 0.0280, unpaired $t$ test. **(D)** GluA1 protein levels as measured by mean optical band density relative to tubulin in empty pSUPER, Ctr shRNA, and Prr7 shRNA-transfected cortical neuron whole-cell extracts, with representative Western blot images. Data = mean normalized to pSUPER empty condition ± SD, n = 3, *$P$ = 0.0105, unpaired $t$ test. **(E, F)** Average whole-cell GluA1 puncta intensity and (F) surface GluA1 puncta number for hippocampal neurons transfected with GFP, with either control (plain bars) or Prr7 shRNA (patterned bars), and treated with PTX or EtOH for 48 h. Data = mean ± SD, n = 3, two-way ANOVA with Tukey's post hoc HSD test. Whole-cell GluA1: Ctr EtOH versus Prr7 EtOH: *$P$ = 0.0228; Ctr EtOH versus Ctr PTX: ns $P$ = 0.0618. Surface GluA1: Ctr EtOH versus Prr7 EtOH: *$P$ = 0.0121; Ctr EtOH versus Ctr PTX: ns $P$ = 0.1193. All scale bars shown = 5 $\mu$m.
Source data are available for this figure.

Finally, we explored whether the observed reductions in spine density and GluA1 surface expression upon Prr7 knockdown translated into corresponding alterations in miniature excitatory postsynaptic currents (mEPSCs) of pyramidal neurons, using whole-cell patch-clamp electrophysiological recordings (Fig S7). Surprisingly, neither mEPSC frequencies nor amplitudes were significantly different between Prr7 shRNA and control-shRNA transfected hippocampal pyramidal neurons (Fig S7A). In contrast, paired-pulse

facilitation (PPF), an indicator of presynaptic function, was significantly elevated in Prr7 knockdown pyramidal neurons (Fig S7B). Consistent with the lack of effect on mEPSC frequency, the density of excitatory synaptic PSD-95/synapsin-1 co-clusters forming onto hippocampal pyramidal neurons was not affected by Prr7 knockdown (Fig S7C). Thus, loss of Prr7 spares postsynaptic function but significantly impacts presynaptic function, possibly involving a non–cell-autonomous mechanism (see the Discussion section for further details).

### miR-329 and miR-495 are required for Prr7 down-regulation by PTX

Next, we explored the mechanisms underlying PTX-dependent down-regulation of Prr7. Following the observations that Prr7 expression is regulated at the RNA level upon PTX treatment, we hypothesized that it may be regulated posttranscriptionally by miRNAs. miRNAs already have strong implications toward activity-dependent synaptic plasticity mechanisms, including HSD (Cohen et al, 2011; Fiore et al, 2014; Rajman et al, 2017).

Upon analysis of the Prr7 3′UTR sequence using the TargetScan algorithm, we found four predicted miRNA-binding sites, two of which overlap with one another (Fig 3A). We examined the effect of inhibiting two of the four miRNA candidates, miR-329-3p and miR-495-3p, on Prr7 mRNA expression in the context of PTX treatment, through the use of a luciferase reporter with the Prr7 3′ UTR cloned downstream of a firefly gene. We selected these two miRNAs for further studies because miR-495-3p is the most abundant of the four candidates, and miR-329-3p has been previously implicated in KCl-dependent dendritogenesis (Fiore et al, 2009). We observed a reduction in firefly luciferase activity upon PTX stimulation, which was prevented by a cocktail of miR-329-3p and miR-495-3p inhibitors (antisense locked nucleic acid inhibitors "pLNAs"). Importantly, this effect was not seen when a reporter with mutated binding sites for these miRNAs on the Prr7 3′ UTR was used (Fig 3B), demonstrating that the effects were mediated by the miRNA-binding sites present in the Prr7 3′UTR. The same effect was observed when transfecting miR-495-3p pLNA alone (Fig 3C). A trend was observed for miR-329-3p pLNA alone, although the effect did not reach statistical significance (Fig 3D). Taken together, these findings indicated that Prr7 is a direct target of miR-329-3p and miR-495-3p during PTX-mediated HSD.

We then asked whether the miR-329-3p and miR-495-3p regulation of Prr7 during downscaling as suggested by luciferase could also be seen at the protein level for dendrite-localized Prr7. Decreases in Prr7 in dendrites upon PTX were prevented when cells were transfected with the cocktail of miR-329-3p and miR-495-3p pLNAs (Fig 3E) but not for miR-329-3p or miR-495-3p pLNA alone, thereby confirming our results from luciferase assays and indicating an additive inhibitory role for miR-329-3p and miR-495-3p in Prr7 regulation.

Because miRNA inhibition appeared to up-regulate Prr7 only in the context of PTX stimulation, we speculated that miR-329-3p and miR-495-3p themselves could be subject to PTX-dependent regulation. We examined endogenous miRNA activity in the mock and PTX-stimulated hippocampal neurons through use of a single-cell dual fluorescence assay ("sensor assay"), as previously described

(Fiore et al, 2009). Specifically, the assay uses polycistronic vectors expressing both GFP and dsRed, whereby dsRed expression is posttranscriptionally controlled by the presence of two perfectly complementary binding sites for the miRNA of interest within the dsRed 3′ UTR (Fig 3F). If miRNAs of interest are active within a given cell, they would bind to the dsRed 3′UTR and down-regulate dsRed expression. Thus, cells expressing only GFP without dsRed were counted as "miRNA positive," and those expressing dsRed were counted as "miRNA negative." Hippocampal neurons were transfected either with a control sensor (containing a sequence nonspecific to any known miRNAs), a miR-329-3p or a miR-495-3p sensor. Subsequently, "miRNA positive" versus "miRNA negative" cells were manually scored over the entirety of each coverslip for all conditions (Fig S8). The proportion of miRNA-positive neurons increased upon PTX treatment for both the miR-329-3p and miR-495-3p sensor transfections (Fig 3G). This induction was not seen when a pLNA against the respective miRNA was co-transfected with the sensor, indicating that the sensor could reliably detect endogenous miRNA activity. In conclusion, PTX treatment increased the proportion of neurons displaying active miR-329-3p and miR-495-3p.

Next, we wanted to test whether the observed PTX-dependent increase in miR-329/495 activity was because of an up-regulation of miRNA expression. qRT-PCR analysis of these two miRNAs in RNA extracts obtained from compartmentalized neuron cultures indicated a nonsignificant PTX-dependent up-regulation in mature miRNA levels for both miR-329-3p and miR-495-3p in the cell body upon PTX treatment (Fig 3H). In the process compartment, only mature miR-495-3p but not miR-329-3p showed nonsignificant increases by PTX (Fig 3I). These findings suggest that in the case of miR-495-3p, PTX-dependent activity increase might involve a local up-regulation of miR-495-3p expression in the dendritic compartment.

### miR-329-3p and miR-495-3p are required for synaptic depression induced by PTX and Prr7 knockdown

We next asked whether miR-329-3p and miR-495-3p were functionally involved in HSD. Therefore, we measured spine density in cells transfected with miR-329-3p and miR-495-3p pLNAs in the presence or absence of PTX treatment (48 h). We found that both miR-329-3p and miR-495-3p inhibition, separately and together, rescued PTX-mediated spine density reduction (Fig 4A). The rescue effect was most pronounced when using a miR-329/495 pLNA cocktail, consistent with our results from Prr7 regulation (Fig 3B and E).

To corroborate Prr7 as an important downstream target in miR329/495–mediated HSD, we further asked whether the impaired HSD induced by the pLNA cocktail could be reinstated by lowering Prr7 levels through co-transfection of Prr7 shRNA. Consistent with this idea, transfection of Prr7 shRNA, but not control shRNA, restored the PTX-induced spine density reduction in the presence of miR-329 and miR-495 pLNAs (Fig 4A and B). This result demonstrates that Prr7 is a key target of miR-329/-495 in PTX-mediated HSD.

We went on to test whether increasing levels of miR-329 and -495 was sufficient to induce spine elimination in the absence of PTX,

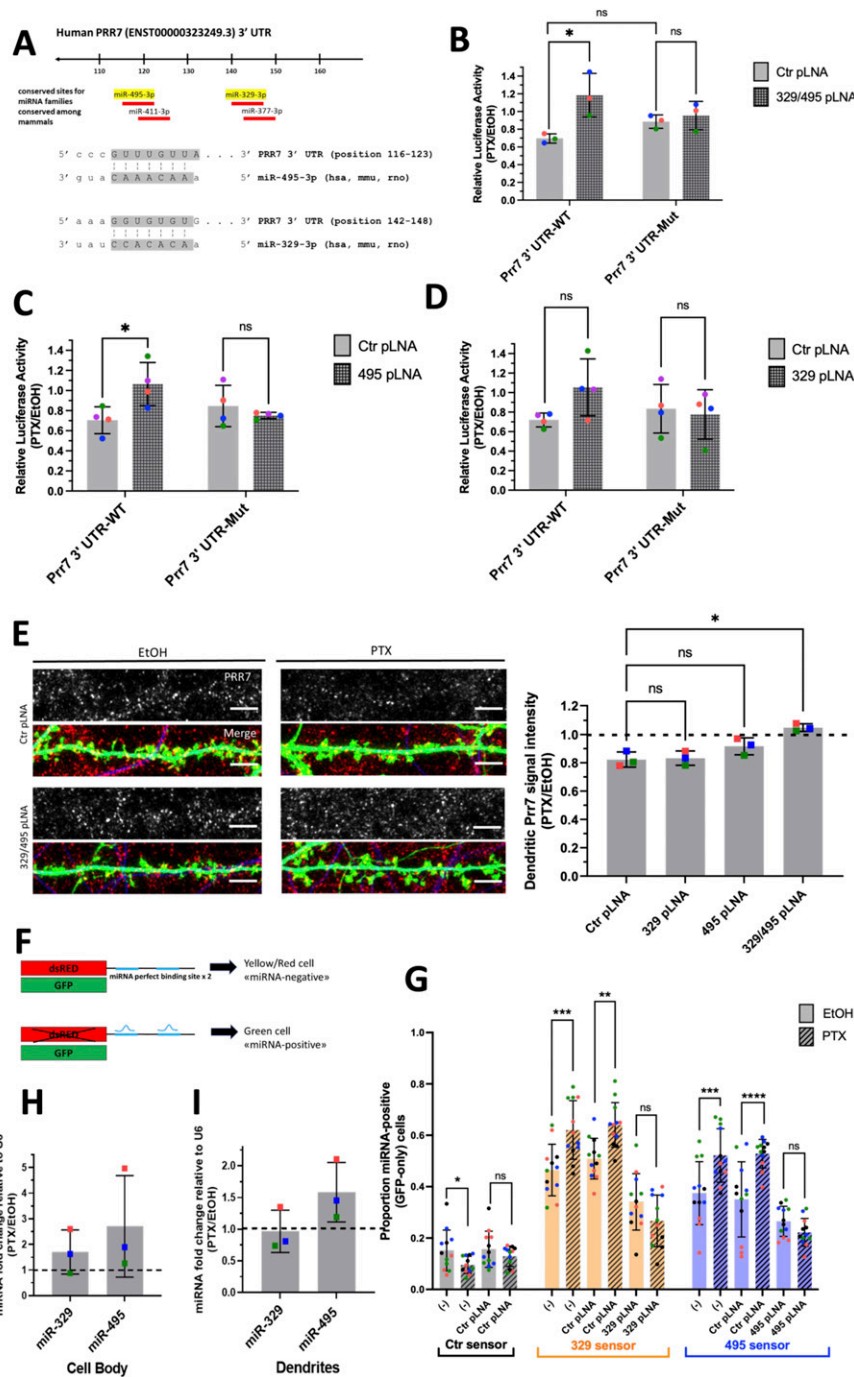

**Figure 3. miR-329 and miR-495 are required for Prr7 down-regulation by PTX.**

**(A)** Predicted miRNA-binding sites for human Prr7 3′ UTR and seed matches for miR-329 and miR-495 from TargetScan (http://www.targetscan.org/). **(B, C, D)** Rat hippocampal neurons were co-transfected at DIV13 with either pmiRGLO Prr7 WT or Mut 3′ UTR plasmids (50 ng) and control pLNA, 329 pLNA, 495 pLNA (20 pmol), or miR-329/495 pLNA mix (10 pmol each) and treated with either EtOH or 100 µM PTX at DIV18. Cells were lysed at DIV20 and firefly/renilla luciferase activity ratios measured. Data = mean normalized to EtOH condition ± SD, n = 3–4, two-way ANOVA with Tukey's post hoc homeostatic synaptic depression test. 329/495 pLNA data: Ctr WT versus 329/495 WT: *P = 0.0194; Ctr WT versus Ctr Mut: ns P = 0.4724; Ctr Mut versus 329/495 Mut: ns P = 0.9452. 495 pLNA data: Ctr WT versus 495 WT: *P = 0.0397; Ctr Mut versus 495 Mut: ns P = 0.8484. 329 pLNA data: Ctr WT versus 329 WT: ns P = 0.2275; Ctr Mut versus 329 Mut: ns P = 0.9841.
**(E)** Representative dendrite images showing Prr7 expression (gray scale, top panels) and merged Prr7 (red), GFP (green), and Map2 (blue) signals (bottom panels) in Ctr or 329/495 pLNA-transfected hippocampal neurons treated with 48 h EtOH or PTX. Scale bars = 5 µm. On the right, average Prr7 punctum intensity in hippocampal cells transfected with GFP (150 ng) and control, miR-329, miR-495 pLNA (20 pmol), or miR-329/495 pLNA mix (10 pmol each) at DIV13 and treated with PTX or EtOH on DIV19 for 48 h. Data = mean normalized to EtOH condition ± SD, n = 3, one-way ANOVA with Tukey's post hoc homeostatic synaptic depression test. Ctr versus 329: ns P = 0.9946; Ctr versus 495: ns P = 0.1739; Ctr versus 329/495: *P = 0.0023.
**(F, G)** Schematic of single-cell dual-fluorescence miRNA sensor assay and measurement of endogenous miR-329 and miR-495 activity upon PTX treatment in hippocampal neurons. Cells were transfected with miR-329, miR-495, or control sensor (125 ng), with or without Ctr, miR-329, or miR-495 pLNA (5 pmol) at DIV13, treated with 100 µM PTX or EtOH at DIV19, and fixed at DIV21. The number of neurons expressing GFP only without dsRed versus those expressing dsRed was counted for three coverslips per condition. Data = proportion of GFP+ cells/total cell count for three coverslips counted per independent experiment ± SD, n = 4, binomial generalized mixed effects model (GLMM) with post hoc tests conducted using the lme4 R and multcomp packages, with P-values adjusted by Bonferroni's method. Ctr sensor data: PTX versus mock: *P = 0.041206; Ctr pLNA PTX versus Ctr pLNA mock: ns P = 0.457119. 329 sensor data: PTX versus mock: ***P = 0.000316; Ctr pLNA PTX versus Ctr pLNA mock: **P = 0.001658; 329 pLNA PTX versus 329 pLNA mock: ns P = 0.05349. 495 sensor data: PTX versus mock: ***P = 0.000145; Ctr pLNA PTX versus Ctr pLNA mock: ****P = 3.78 × 10⁻⁶; 495 pLNA PTX versus 495 pLNA mock: ns P = 0.23609. **(H, I)** Mature miR-329 and miR-495 levels in cell body and dendrite compartments of hippocampal neurons treated with either EtOH or PTX for 48 h. Data = mean normalized to EtOH condition ± SD, n = 3, one-sample t test with hypothetical mean set to 1. Cell body:

miR-329 ns P = 0.2923, miR-495 ns P = 0.2742: dendrites: miR-329 ns P = 0.8777, miR-495 ns P = 0.1641. Source data are available for this figure.

thereby mimicking HSD. Toward this end, we constructed chimeric miR-329 and -495 overexpressing plasmids using a previously described strategy (Christensen et al, 2010; Fig S9A). We observed an approximately twofold overexpression of the miRNA of interest relative to control (Fig S9B), accompanied by consistently downward trends of Prr7 in dendrites (Fig S10A and B). Despite the moderate effects on

miRNA overexpression and Prr7 reduction achieved with this approach, stable overexpression of miR-329 was sufficient to induce a significant reduction in spine density and miR-495 overexpression led to a consistent downward trend (Fig 4C and D). Thus, miR-329/495 overexpression mimics PTX-induced miR-329/495 expression followed by spine elimination in transfected hippocampal neurons.

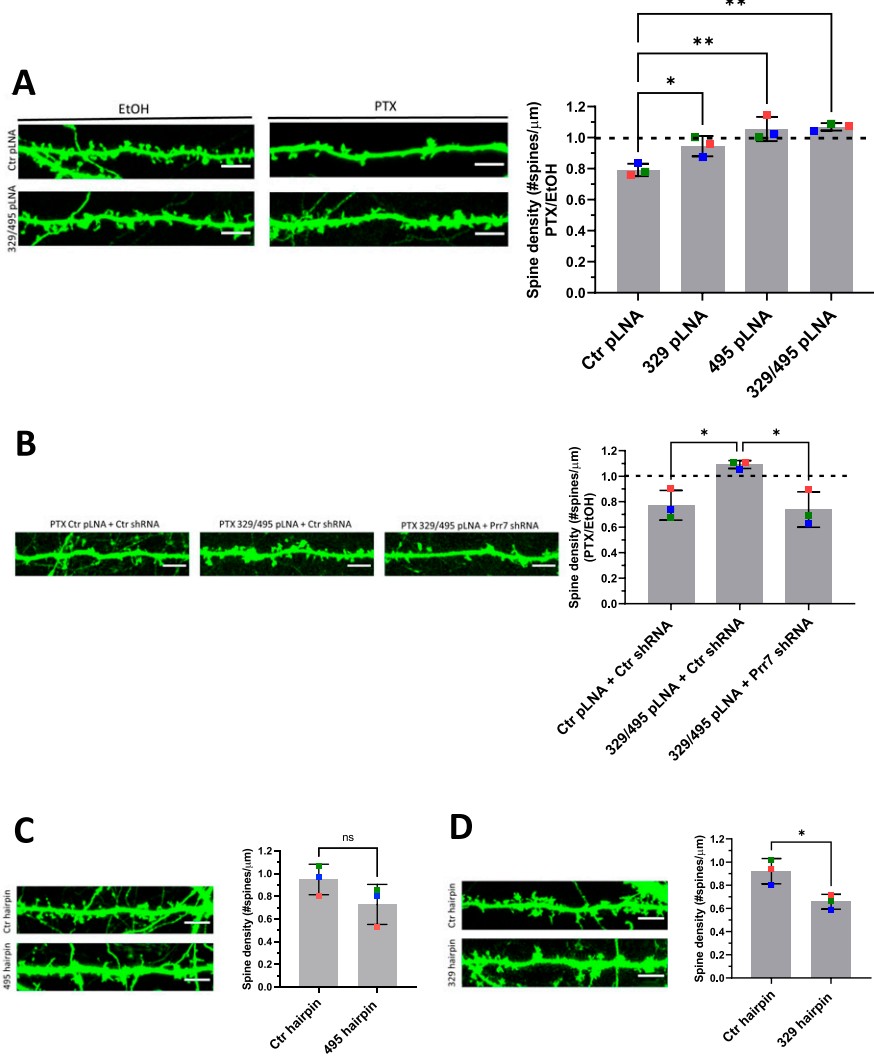

**Figure 4. miR-329– and miR-495–mediated Prr7 down-regulation are required for synaptic depression induced by PTX.**
**(A)** Spine densities of hippocampal cells transfected with GFP (150 ng) and control, miR-329, miR-495 pLNA (20 pmol), or miR-329/495 pLNA mix (10 pmol each) on DIV13, treated with EtOH or 100 µM PTX on DIV19 for 48 h. Representative GFP images showing dendrites for each condition are shown. Ctr versus 329: *P = 0.0400; Ctr versus 495: **P = 0.0018; Ctr versus 329/495: **P = 0.0013. **(B)** Spine densities of hippocampal cells transfected with GFP (150 ng), control pLNA (20 pmol), or miR-329/495 pLNA mix (10 pmol each) and control or Prr7 shRNA (2.5 ng pSUPER) on DIV13, treated with EtOH or PTX on DIV19 for 48 h (Ctr + Ctrsh versus 329/495 + Ctrsh: *P = 0.0240; 329/495 + Ctrsh versus 329/495 + Prr7sh: *P = 0.0155). For these pLNA data, data = mean normalized to EtOH condition ± SD, n = 3, one-way ANOVA with Tukey's post hoc homeostatic synaptic depression test. **(C, D)** Spine densities of hippocampal cells transfected with control or (C) miR30a-495 chimeric hairpin (500 ng), (D) miR30a-329 chimeric hairpin (500 ng) on DIV13 and fixed on DIV18-19 (329 hp) or DIV21 (495 hp). Data = mean ± SD, n = 3, unpaired t test (Ctr versus 495: ns P = 0.1608; Ctr versus 329: *P = 0.0223). Representative GFP images showing dendrites for each condition are shown.
Source data are available for this figure.

## SPAR/CDK5 pathway is downstream of miR-329/miR-495/Prr7 in HSD

We further explored the pathway downstream of miR329/495/Prr7 which mediates the effects on spine density. One attractive candidate is the Plk2/SPAR pathway which has previously been implicated in HSD (Pak & Sheng, 2003). Specifically, Plk2-mediated phosphorylation of SPAR is followed by proteasome-dependent SPAR degradation, leading to excitatory synapse weakening and spine loss. Intriguingly, both SPAR and Prr7 have been shown to interact with PSD-95, an important scaffold protein required for the integrity of the postsynaptic density. We therefore speculated that Prr7 might protect SPAR from Plk2-mediated degradation, possibly in conjunction with PSD-95. Thus, we tested whether reducing Prr7 levels affected SPAR expression in a way consistent with a role in HSD. We found a reduction in SPAR levels in cortical neurons nucleofected with Prr7 shRNA through Western blotting (Fig 5A). Moreover, dendrite-localized SPAR protein was reduced in Prr7 shRNA-transfected hippocampal neurons (Figs 5B and S11). SPAR

reduction was also seen upon PTX treatment as previously reported (Pak & Sheng, 2003). Therefore, Prr7 may stabilize SPAR at basal levels of network activity.

CDK5 kinase-mediated SPAR phosphorylation primes SPAR for targeting by Plk2 (Seeburg et al, 2008). To determine whether Prr7 serves to stabilize SPAR by interfering with CDK5 activity, we treated Prr7 shRNA-transfected cells with 10 µM roscovitine, a CDK5 inhibitor, and quantified SPAR puncta intensity in dendrites. We found that a reduction in SPAR protein levels was no longer seen in Prr7 shRNA-transfected cells treated with roscovitine relative to DMSO (Fig 5C). Moreover, roscovitine treatment rescued the spine density reduction in the Prr7 knockdown condition (Fig 5C). Because roscovitine is a broad-spectrum inhibitor that targets several members of the Cdk family and is not specific to CDK5, we further quantified SPAR expression and spine density in cells cotransfected with Prr7 shRNA and a previously published dominant-negative CDK5 construct (CDK5D144N, Seeburg et al, 2008). We found that CDK5D144N elevated neuronal SPAR protein expression in a dose-dependent manner (Fig S12A and B) and prevented reduced

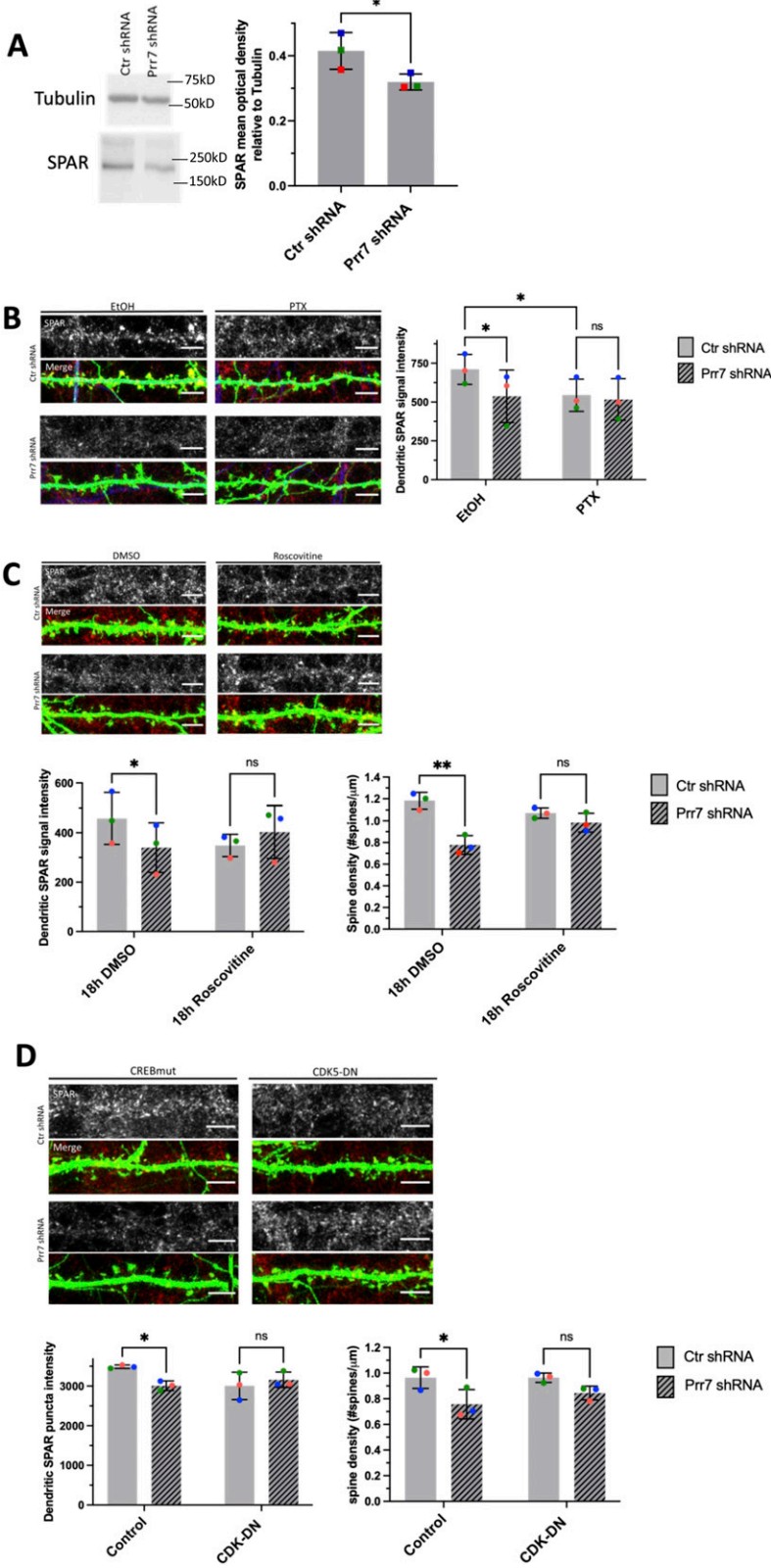

**Figure 5.  SPAR/CDK5 pathway is downstream of miR-329/495/Prr7 regulation in homeostatic synaptic depression (HSD).**

**(A)** SPAR protein levels relative to tubulin in rat cortical neurons nucleofected with control or Prr7 shRNA (2 μg) on day of dissociation (E18) and harvested for protein extraction 5 d later, with representative Western blot. Data = mean ± SD, n = 3, ns *P* = 0.0541, unpaired *t* test. **(B)** Representative dendrite images showing SPAR expression (gray scale, top panels) and merged SPAR (red), GFP (green), Map2 (blue) signals (bottom panels) in Ctr or Prr7 shRNA-transfected hippocampal neurons treated with 48 h EtOH or PTX. To the right, average SPAR punctum intensity in dendrite selection in rat hippocampal cells transfected with GFP (150 ng), control or Prr7 shRNA (7.5 ng pSUPER) at DIV13, treated with either EtOH (1:500 volume) or 100 μM PTX at DIV19 for 48 h and then immunostained for SPAR. Data = mean ± SD, where each point represents grand average in dendrites for the 7–10 cells imaged in a single experiment. n = 3, two-way ANOVA with Tukey's HSD post hoc test. Ctr EtOH versus Prr7 EtOH: *P* = 0.0429; Ctr EtOH versus Ctr PTX: *P* = 0.0462; Ctr PTX versus Prr7 PTX: ns *P* = 0.6932. **(C)** Representative dendrite images showing SPAR expression (gray scale, top panels) and merged SPAR (red) and GFP (green) signals (bottom panels) in Ctr or Prr7 shRNA-transfected hippocampal neurons treated with 18 h DMSO or roscovitine. Average dendrite SPAR punctum intensity in rat hippocampal cells transfected with GFP (150 ng) and either control or Prr7 shRNA (7.5 ng) at DIV13, treated with either DMSO (1:1,000 volume) or roscovitine (10 μM) at DIV19 for 18 h and then immunostained for SPAR. Data = mean ± SD, where each point represents grand average in dendrites for the 8–12 cells imaged in a single experiment. n = 3. Ctr DMSO versus Prr7 DMSO: *P* = 0.0266; Ctr Ros versus Prr7 Ros: ns *P* = 0.2328. Spine densities for these same cells were measured. Ctr DMSO versus Prr7 DMSO: **P* = 0.0064; Ctr Ros versus Prr7 Ros: ns *P* = 0.4288. For these roscovitine data, data = mean ± SD, two-way ANOVA with Tukey's HSD post hoc test. **(D)** Representative dendrite images showing SPAR expression and merged SPAR (red) and GFP (green) signals in Ctr or Prr7 shRNA-transfected neurons, with further co-transfections with either inactive form of CREB (CREB-VP16m labeled "Control," 400 ng) or CDK5-dominant negative (CDK-DN, 400 ng) and then immunostained for SPAR. Data = mean ± SD, where each point represents grand average in dendrites for the 10 cells imaged in a single experiment. n = 3, two-way ANOVA with Tukey's HSD post hoc test. Ctr shRNA control versus Prr7 shRNA control: **P* = 0.0223; Ctr shRNA CDK-DN versus Prr7 shRNA CDK5-DN: ns *P* = 0.1814. Spine densities for these same cells were measured. Ctr shRNA control versus Prr7 shRNA control: **P* = 0.0469; Ctr shRNA CDK-DN versus Prr7 shRNA CDK5-DN: ns *P* = 0.3140. Scale bars = 5 μm. Source data are available for this figure.

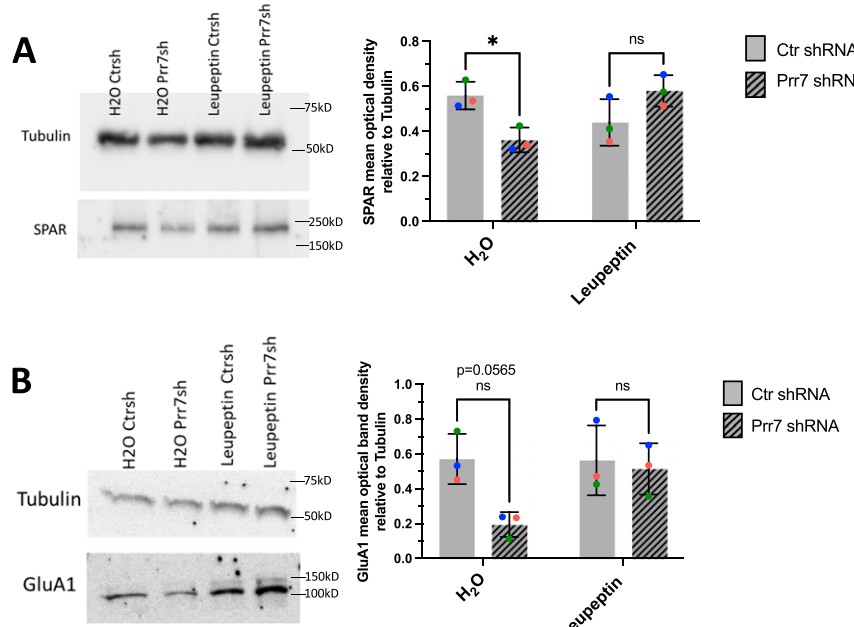

**Figure 6.   Prr7 impedes SPAR and GluA1 degradation pathways.**

**(A, B)** SPAR and (B) GluA1 protein levels relative to tubulin in rat cortical neurons nucleofected with control or Prr7 shRNA (2 $\mu$g) on day of dissociation (E18), treated with either leupeptin (200 $\mu$g/ml) or equivalent volume of water for 20–21 h on DIV5, and then harvested on DIV6, with representative Western blot. Data = mean ± SD, n = 3, two-way ANOVA with Tukey's homeostatic synaptic depression post hoc test. SPAR Western: $H_2O$ Ctr shRNA versus $H_2O$ Prr7 shRNA: *$P$ = 0.0477; leupeptin Ctr shRNA versus leupeptin Prr7 shRNA: ns $P$ = 0.1783. GluA1 Western: $H_2O$ Ctr shRNA versus $H_2O$ Prr7 shRNA: ns $P$ = 0.0565; leupeptin Ctr shRNA versus leupeptin Prr7 shRNA: ns $P$ = 0.9755.
Source data are available for this figure.

SPAR expression upon Prr7 shRNA transfection (Fig 5D, lower left panel). Furthermore, the Prr7 knockdown-mediated spine density reduction was strongly attenuated by CDK5D144N co-expression (Fig 5D, lower right panel). These findings suggest that Prr7 functions upstream of CDK5, potentially stabilizing dendritic spines through protecting SPAR from CDK5-mediated priming phosphorylation, which in turn is required for SPAR phosphorylation by Plk2 and subsequent degradation.

To directly address a role for SPAR degradation downstream of Prr7, we performed Prr7 knockdown in the presence of the proteasomal inhibitor MG-132 and the broad-spectrum serine, cysteine, and threonine inhibitor leupeptin. Although MG-132 was toxic to rat hippocampal neurons even at low concentrations (data not shown), leupeptin was well tolerated and efficiently prevented the reduction in SPAR protein levels induced by Prr7 knockdown (Fig 6A). Because GluA1 reduction was also induced by Prr7 shRNA, we additionally examined the effect of leupeptin treatment on GluA1 and found a prevention of the GluA1 loss as well (Fig 6B). Together, these observations indicated that Prr7 protects both SPAR and GluA1 proteins by interfering with their respective protein degradation pathways.

## Discussion

Our study demonstrates the requirement of miR-329– and miR-495–mediated down-regulation of Prr7 underlying dendritic spine elimination in HSD. From our results, we present the following model (Fig 7). Under basal conditions, Prr7 mRNA is actively translated as the targeting of the Prr7 3′ UTR by miRNAs is inhibited by a yet unknown mechanism. Prr7 protein is required for the stabilization of SPAR through inhibiting the activity of CDK5, thereby

maintaining the integrity of the postsynaptic density, including the stabilization of GluA1-containing AMPARs at the surface.

In contrast, after chronic activity, miR-329 and miR-495 are activated, and miR-495 expression is increased in dendrites. These miRNAs repress translation of Prr7 mRNA. In the absence of Prr7 protein, CDK5 phosphorylates SPAR, leading to an association between SPAR and Plk2 and subsequent SPAR degradation. The loss of SPAR results in the destabilization of PSD-95 complexes and GluA1 degradation, ultimately resulting in spine elimination.

### Role of Prr7 in synaptogenesis and plasticity

Through Prr7 knockdown studies, we have revealed that Prr7 reduction leads to a decrease in the spine number (Fig 2A) and GluA1 protein levels (Fig 2D–F), recapitulating two hallmarks of HSD. Although Prr7 was not found to associate with AMPARs in a previous study through immunoprecipitations (Kravchick et al, 2016), it is still possible that Prr7 influences AMPAR dynamics indirectly through interaction with other PSD components, for example, the AMPAR auxiliary subunit Stargazin, which binds to both AMPARs and PSD-95 (Bats et al, 2007). Nevertheless, the current results are in agreement with the previously presented idea that Prr7 reduction serves a neuroprotective function against over-excitation (Kravchick et al, 2016) and Prr7 forms part of the postsynaptic density core to promote neuronal maturation (Murata et al, 2005). The direct and indirect protein interactions involving Prr7 and Prr7-associated complexes formed under basal versus stimulated conditions need further clarification.

Our results using sparse transfection of hippocampal neuron cultures clearly indicate a cell-autonomous, postsynaptic function of Prr7, at least at the level of spine morphogenesis. In contrast, a non–cell-autonomous function of Prr7 through exosomal secretion

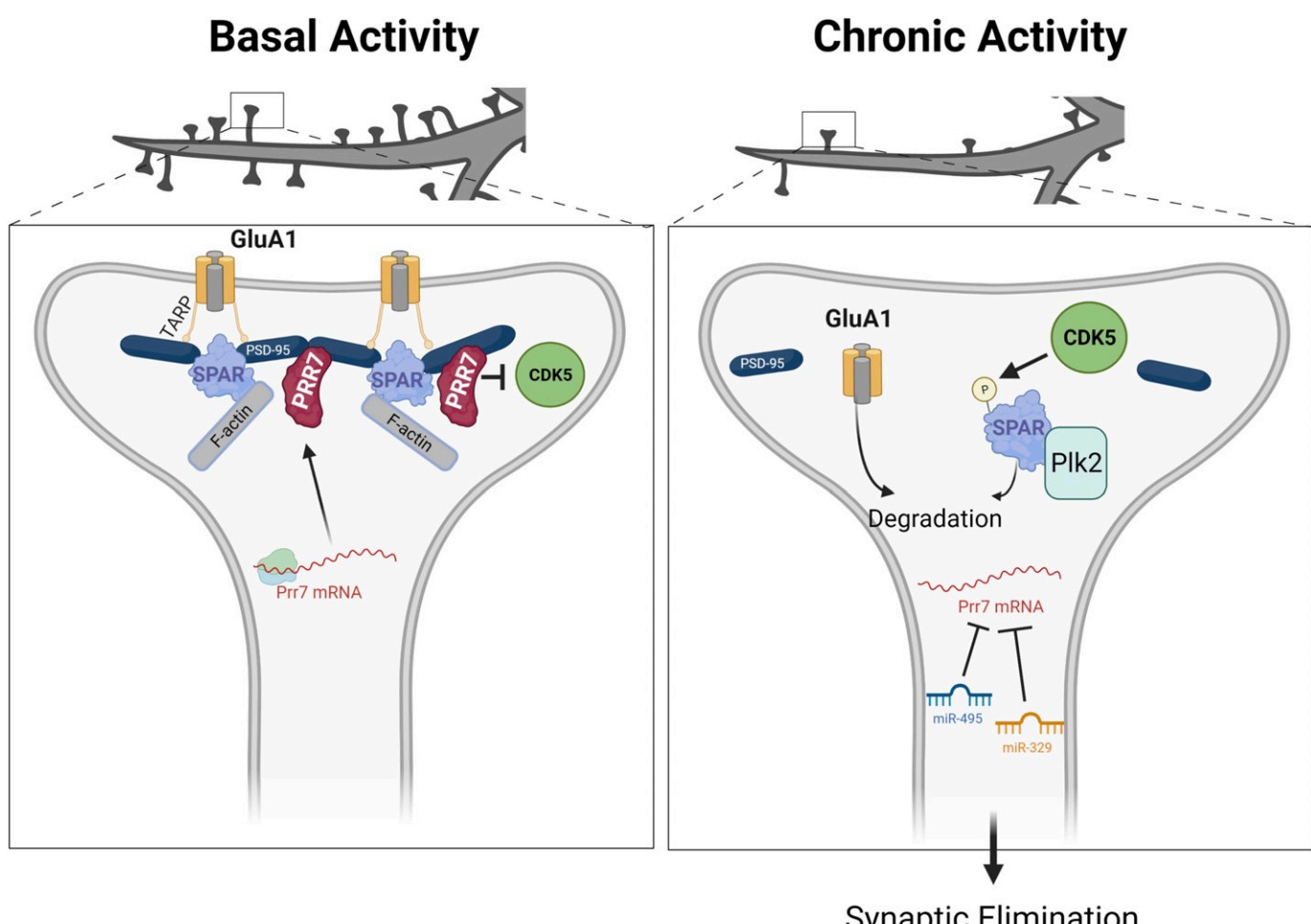

**Figure 7. Model of miRNA-mediated Prr7 down-regulation and downstream effects on SPAR during homeostatic synaptic depression.**
Left panel: under basal conditions, Prr7 mRNA is actively translated as the miR-329 and miR-495 activity are inhibited (by a yet unknown mechanism). Prr7 protein stabilizes SPAR through inhibiting CDK5 activity and preventing SPAR phosphorylation. Thereby, the integrity of the postsynaptic density is maintained. Right panel: after chronic activity, miR-329 and miR-495 are activated, and miR-495 expression specifically is increased in dendrites. These miRNAs inhibit Prr7 mRNA, and thus, Prr7 protein is lost. CDK5 phosphorylates SPAR, leading to targeting of SPAR by Plk2 and subsequent degradation. As a result of SPAR loss, PSD-95 complexes are destabilized and GluA1 is degraded, leading to elimination of spines. (Figure generated with BioRender).

and Wnt inhibition has previously been reported (Lee et al, 2018). In this study, treatment of exosomal Prr7-rich supernatant in hippocampal cultures led to a reduction of glutamatergic synapses, which is contrary to our observations. The differences in incubation time between treatment and imaging (fixation 18 h versus 8 d post-transfection) may account for this discrepancy. Namely, it is possible that upon acute (18 h) Prr7 overexpression, spines are eliminated because of a rapid exosomal Prr7 secretion from the soma. In contrast, over a time scale of days, Prr7 might accumulate in the synapto-dendritic compartment where it promotes synaptogenesis to compensate for the initial spine loss. In fact, our observation that paired-pulse ratios (PPRs), a classical indicator of presynaptic activity, are increased in Prr7 knockdown neurons also suggests the involvement of a non cell-autonomous mechanism, possibly Prr7 exosomal secretion, in our model. According to this, lack of exosomal uptake of Prr7 into the presynaptic neuron through a trans-synaptic signaling mechanism might lead to increased presynaptic CDK5 activity, greater phosphorylation of

Synapsin1 and sequestration. This in turn might be responsible for a decreased vesicle trafficking to the active zone and the observed PPR increase. In addition, it is also noteworthy that the observed increase in excitatory synapses upon Prr7 knockdown in Lee et al (2018) was solely based on PSD-95 puncta number and that effects on spines were not directly addressed.

Surprisingly, the robust reduction in spine density in Prr7-shRNA transfected hippocampal pyramidal neurons was neither paralleled by a corresponding decrease in mEPSC frequency nor in excitatory synaptic PSD-95/Synapsin-1 co-clusters. Several mechanisms could explain such a dissociation of morphological and electrophysiological parameters. First, Prr7 knockdown might lead to a specific elimination of spines which lack presynaptic input, possibly immature spines. Second, the loss of spine-associated functional synapses in Prr7 knockdown neurons might be compensated by an increase in synapses forming onto the dendritic shaft and/or the neuronal soma. Finally, electrophysiological changes in HSD might be mainly because of a reduction in GluA1-

lacking AMPARs, which are mostly spared in Prr7 knockdown neurons. In the future, we will perform a more detailed characterization of the subcellular localization of synaptic co-clusters and AMPAR complexes in dendrites to clarify the involved mechanisms.

We further elucidated a mechanism in which CDK5/SPAR is controlled downstream of Prr7 activity (Fig 5A–D). Our results that the broad-spectrum protease inhibitor leupeptin interferes with SPAR degradation are somehow at odds with a previous study (Pak & Sheng, 2003) which reported that SPAR degradation after Plk2 targeting is prevented with the proteasome inhibitor MG-132 but not leupeptin. However, Pak and Sheng (2003) used a lower leupeptin concentration and conducted experiments with recombinant SPAR in COS cells which do not express CDK5 or Prr7. Thus, a contribution of lysosome activity to SPAR degradation in neurons cannot be ruled out. In this regard, it was shown that lysosomes participate in the activity-dependent degradation of GluA1-containing AMPARs (Schwarz et al, 2010). On the other hand, leupeptin displays partial activity toward the proteasome due its inhibitory effect on threonine proteases (Harer et al, 2012), leaving open the possibility that at least part of the effects on SPAR degradation we observe are dependent on proteasome activity.

It is interesting to consider this new pathway linking miR-329/495/Prr7 to CDK5/SPAR in relation to a study describing miR-134-dependent SPAR regulation via Pum2 down-regulation and Plk2 function (Fiore et al, 2014). It is known that upon chronic activity, Plk2 is activated, and there is a bifurcation into two downstream branches (Seeburg et al, 2008; Seeburg & Sheng, 2008; Evers et al, 2010): (1) activated Plk2 phosphorylates SPAR, leading to GluA1/GluA2 internalization, and (2) activated Plk2 phosphorylates the GluA2-interacting protein NSF, promoting specifically GluA2 internalization. Intriguingly, in Fiore et al's study, it was found that the miR-134 pathway only affected GluA2 levels, and therefore, it was suggested to connect only to the second branch. Considering Prr7 knockdown affected GluA1 expression (Fig 2D–F), it would be plausible that conversely miR-329/495/Prr7 feeds into specifically the first branch via influencing CDK5.

In addition, in light of numerous studies highlighting the role of CDK5 on synaptic plasticity (mainly concerning memory function), it is important to consider the implications of Prr7-mediated CDK5 inhibition on the general regulation of synapse number and function. CDK5 promotes synaptic weakening by phosphorylating multiple targets at both pre- and postsynaptic levels. Namely, CDK5, upon activation by cofactor p35, phosphorylates and down-regulates voltage-gated calcium channels (Tomizawa et al, 2002) and sequesters Synapsin1 (Verstegen et al, 2014), thereby inhibiting neurotransmitter release. A number of postsynaptic targets of CDK5 have been identified, including NMDA receptor subunit NR2B (Hawasli et al, 2007; Plattner et al, 2014), PSD95 (Morabito et al, 2004), Liprinα1 (Huang et al, 2017), and DARPP-32 (Bibb et al, 1999; Brito et al, 2019), whose CDK5-mediated down-regulation lead to loss of dendritic spines and reduced neurotransmission. Therefore, given such widespread targets of CDK5, modulation of Prr7 in conditions of CDK5 aberrancy may serve as a strategy to restore synaptic function and correct for the dysregulation of multiple synaptic proteins downstream of CDK5 activity beyond SPAR.

## Role of miRNAs in local activity-dependent regulation of Prr7 during HSD

We have demonstrated that Prr7 expression at both RNA and protein levels is consistently reduced in the dendritic compartment in response to chronic activity (Fig 1B–D). The decrease in Prr7 mRNA in dendrites of PTX-treated neurons was also observed from RNA-seq analyses (Colameo et al, 2021). The same RNA-seq dataset revealed enrichment of Prr7 mRNA in dendrites under basal conditions, which would suggest an active dendritic transport mechanism for Prr7 mRNA. However, it is unlikely that PTX-mediated Prr7 mRNA changes are solely because of a redistribution of Prr7 mRNA by active transport given our observation of a similar down-regulation, albeit to a lesser extent, in cell bodies upon PTX (Fig 1B and D). Furthermore, based on our previously published RNA-seq dataset (Colameo et al, 2021), we have observed that Prr7 pre-mRNA reads were comparable in the cell body compartment between PTX- and mock-treated neurons, which strongly argues against an important contribution of transcriptional inhibition to the observed reductions in Prr7 mRNA levels.

Together, these observations support the idea that there is local regulation of Prr7 in the synapto-dendritic compartment during HSD. In this regard, local homeostatic mechanisms at the level of individual dendritic domains (Ju et al, 2004; Sutton et al, 2006) and at individual synapses (Hou et al, 2008) have been previously demonstrated. The exact mechanism by which local down-regulation of Prr7 occurs is yet to be uncovered and will need the employment of techniques that allow the visualization of reductions in newly synthesized proteins. These include for example puromycin labeling with proximity ligation assay (puro-PLA) (Dieck et al, 2015) or single-molecule imaging of nascent peptides combined with single-molecule FISH, as performed in hippocampal dendrites (Wu et al, 2016).

We have shown that miR-329 and miR-495 activity and subsequent targeting of the Prr7 3′ UTR are required for Prr7 reduction in dendrites during HSD. Our findings from pLNA experiments are most consistent with an additive repressive effect of these two miRNAs on Prr7 mRNA translation. Such additive effects of multiple miRNAs binding to the same target have been demonstrated previously, for example, with N-cadherin (Rago et al, 2014).

The activity dependency of the miRNA-Prr7 interaction is evidenced by the induction of sensor activity for both miRNAs upon PTX treatment (Fig 3G) and the pLNAs showing effects exclusively under stimulated conditions. However, the mechanisms leading to miR-329 and miR-495 induction appear to be different. In the case of miR-495, mature levels increase, pointing to a PTX-dependent regulation of miR-495 expression (Fig 3I). Because this increase is preferentially observed in dendrites, it might involve increased local miRNA processing, miR-495 transport into dendrites, and/or the local inhibition of miRNA degradation.

With respect to miR-329, the lack of a clear induction in mature miRNA levels suggests mechanisms at the level of the miR-329 RISC, for example, interference of miR-329 RISC binding to the Prr7 3′UTR by an RNA-binding protein which is removed upon PTX treatment. Examples for activity-dependent miRNA-RBP interplay have been previously reported (Kedde et al, 2007; Edbauer et al, 2010; Tominaga et al, 2011; Rajman et al, 2017). In our example, the miR-329

binding site within the Prr7 3′UTR contains a motif which is recognized by the CELF family of RBPs (Ray et al, 2013). Because the expression of several CELF family members is reduced by PTX (Rajman et al, 2017), one can speculate that miR-329 becomes dominant over CELF in PTX-treated neurons, displacing stabilizing CELF from Prr7 and inducing its posttranscriptional silencing.

In addition, the understanding that miR-134, miR-329 and miR-495 activity all lead to SPAR down-regulation in HSD is intriguing, given that these three miRNAs are derived from the same genomic region, termed the miR-379-410 cluster located within the imprinted DLK1-DIO3 region on chromosome 14q32 in humans (da Rocha et al, 2008). Another cluster member, miR-485, also plays a role in homeostatic plasticity through expression regulation of presynaptic synaptic vesicle protein (SV2A) (Cohen et al, 2011). This shared origin of cluster miRNAs not only further support the functional significance of miR-379-410 members in activity-dependent synaptic plasticity mechanisms as previously described (Fiore et al, 2009) but also would point toward an interesting idea that individual cluster members act in distinct yet converging pathways in HSD.

### (Patho)physiological impact of the miR329/495/Prr7 pathway

A previous study (Lackinger et al, 2019) revealed that mice with a constitutive functional deletion of miR-379-410 exhibited heightened sociability and anxiety, along with increased excitatory transmission in hippocampal excitatory neurons and up-regulation of Prr7. These findings not only are consistent with the proposed role of Prr7 in excitatory synaptogenesis but also point toward the connection of miRNA/Prr7 interactions to social or anxiety behavior. In other words, our current results would prompt behavioral studies examining miR-329/495/Prr7 excitatory synapse regulation in vivo. Prr7 knockout mice have been generated in previous studies with no lethal effects in the context of immune regulation (Hrdinka et al, 2016), making the study of hippocampal excitatory transmission and behavior in these mice in the context of miRNA manipulation possible.

Together with the reported involvement of Prr7 in apoptosis (Kravchick et al, 2016), our results may also suggest the importance of miRNA-dependent Prr7 down-regulation in synaptic homeostasis and neuronal survival in the face of excitotoxic insult. Namely, Prr7 knockdown was shown to attenuate the excitotoxic response in hippocampal neurons after NMDAR stimulation by glutamate in a c-Jun–dependent manner (Kravchick et al, 2016). Consistent with this idea, we have shown that Prr7 reduction is necessary for excitatory synapse depression upon chronic stimulation. Given our findings of miR-329– and -495–mediated Prr7 inhibition by PTX, it would therefore be reasonable to ask if these same miRNAs are activated upon glutamate stimulation and if such activation may have neuroprotective effects against excitotoxicity. Taken together, a possible model emerges in which excessive NMDAR stimulation activates miRNAs that target Prr7, thereby reducing synapse-localized Prr7 and preventing Prr7 translocation to the nucleus. Consequently, the absence of dendritic Prr7 leads to spine elimination via SPAR degradation for the purpose of homeostasis, whereas the inhibition of nuclear Prr7 accumulation leads to c-Jun degradation for the purpose of neuronal survival.

More broadly, this idea may be tested in an in vivo context with possible future applications toward neuroprotection after status epilepticus or ischemic stroke as NMDAR overstimulation is implicated in these conditions (McDonough & Shih, 1997; Marshall et al, 2003). In addition, dendritic spine loss has been observed in epilepsy (Swann et al, 2000) and after stroke (Brown et al, 2008), which could indeed suggest the initiation of HSD (in addition to other neuroprotective mechanisms) to counter excitotoxicity. The therapeutic effect of miR-329/-495 administration or Prr7 inhibition in the context of these conditions is yet to be uncovered.

# Materials and Methods

### DNA constructs

All primer sequences used for cloning are indicated in Table S1. miR-30a-chimeric hairpins for miR-329 and miR-495 stable over-expression were generated via polynucleotide cloning into the 3′ UTR of eGFP in the pAAV-hSyn-EGFP vector (Plasmid #114213; Addgene) using BsrGI and HindIII sites, as described previously (Christensen et al, 2010).

Control and Prr7 shRNA vectors were constructed using the pSUPER RNAi System (Oligoengine). Custom primers were designed for polynucleotide cloning into the pSUPER basic vector (VEC-PBS-0001/0002) using BglII/HindIII sites, to generate an shRNA targeting a 19-nucleotide sequence unique to the rat Prr7 coding region (cggaatcggacatgtctaa).

For the Prr7 expression construct, full-length rat Prr7 coding sequence (NM_001109116.1) was amplified from hippocampal rat cDNA and then subcloned into the CMV-pcDNA3 vector using BamHI/XbaI sites. Subsequently, a start codon (atg) with HA-tag (tacccatacgacgtcccagactacgct) was inserted at the HindIII site upstream of Prr7 by polynucleotide cloning. To generate the shRNA-resistant construct, six point mutations in the Prr7 coding region were introduced, such that Prr7 shRNA could no longer recognize the mRNA product, yet the amino acid sequence of the resultant exogenous Prr7 protein (AESDMSK) would remain unchanged. Mutagenesis was performed using Phusion-site directed mutagenesis kit (Thermo Fisher Scientific).

Bi-cistronic reporter constructs for miRNA activity were described previously (Fiore et al, 2009). Two perfectly complementary binding sites for either miR-329 and miR-495, separated by a two-nucleotide linker, were inserted into the dsRED 3′ UTR of the pTracer-CMV-dsRED vector, using XbaI/NotI sites by polynucleotide cloning.

Wild-type and mutant Prr7 3′ UTR luciferase constructs were described and generated previously Lackinger et al (2019), wherein the 3′ UTR of Prr7 (NM_001030296.4) was amplified from mouse DNA and cloned into the pmiRGLO dual-luciferase expression vector (Promega). Mutations in miRNA-binding sites conserved across mammals were introduced by site-directed mutagenesis using Pfu Plus! DNA Polymerase (Roboklon).

CDK5-dominant negative construct (CDK5D144N) previously published (Seeburg et al, 2008) was a kind gift from D. T. Pak. Preliminary validation, and titrations were performed to determine

the optimal vector concentration before experiments (Fig S12). As a control for the CDK5-dominant negative condition, a plasmid for exogenous expression of an inactive form of CREB (CREB-VP16m) was used (gift from ME Greenberg, used in Fiore et al [2014]).

## Cell culture

Primary cortical and hippocampal neuronal cultures were prepared from embryonic day 18 (E18) male and female Sprague-Dawley rats (Janvier Laboratories) as previously described (Schratt et al, 2006). Euthanasia of pregnant rats for the removal of embryonic brains was approved by the Veterinary Office of the Canton Zurich, Switzerland, under license ZH027/2021. Dissociated cortical neurons were directly seeded on six-well plates coated with poly-L-ornithine (used for nucleofections), whereas hippocampal neurons were seeded on poly-L-lysine/laminin-coated coverslips in 24-well plates.

For compartmentalized cell cultures, dissociated hippocampal cells were plated onto 1-μm pore and 30-mm diameter polyethylene tetra-phthalate (PET) membrane filter inserts (Millipore) that were matrix-coated with poly-L-lysine (Sigma-Aldrich) and laminin (BD Biosciences) on the top and bottom, also as described previously (Bicker et al, 2013). With the exception of cells for electrophysiology, all neuron cultures were maintained in Neurobasal-plus (A3582901; Thermo Fisher Scientific) media supplemented with 2% B27, 2 mM GlutaMAX, 100 μg/ml streptomycin, and 100 U/ml penicillin (Gibco; Invitrogen) in an incubator with 5% $CO_2$ at 37°C. Hippocampal cells used for electrophysiology were maintained in Neurobasal-A (10888022; Thermo Fisher Scientific) media supplemented with 2% B27, 2 mM GlutaMAX, 100 μg/ml streptomycin, and 100 U/ml penicillin (Gibco; Invitrogen).

HEK293T cells (Sigma-Aldrich) were maintained in 6-cm dishes in DMEM media containing 10% fetal bovine serum, 1 mM glutamine, 100 U/ml penicillin, and 100 μg/ml streptomycin ("HEK media") in an incubator with 5% $CO_2$ at 37°C.

## Transfections and nucleofections

All transfections of hippocampal cells were performed using Lipofectamine 2000 (Invitrogen), in triplicate wells on DIV13 in Neurobasal plus medium (A3582901; Thermo Fisher Scientific), with the exception of electrophysiology experiments. 1 μg of total DNA was transfected per well in a 24-well plate, where an empty pcDNA3 vector was used to make up the total amount of DNA. Neurons were transfected in Neurobasal plus media in the absence of streptomycin and penicillin for 2 h, replaced with neuron culture media containing ApV (1:1,000) for 45 min, which was washed out and replaced with conditioning media.

Hippocampal neuron transfections for electrophysiology were performed using Lipofectamine 2000 in six replicate wells on DIV9-12 in Neurobasal-A medium (10888022; Thermo Fisher Scientific). 1 μg of total DNA was transfected per well in a 24-well plate, where an empty pcDNA3 vector was used to make up the total amount of DNA. Before addition of the lipofectamine-DNA mix, cells were equilibrated in warm Neurobasal-A containing ApV (1:1,000) without penicillin and streptomycin for 30 min in 37°C. Transfection incubation time was shortened to 1.5 h and in the presence of ApV. Cells were subsequently washed with Neurobasal-A supplemented with

ApV, replaced with conditioning media, and maintained until the day of recording.

Nucleofections were done on cortical neurons using the P3 Primary Cell 4D-Nucleofector X Kit (LZ-V4XP-3024; Lonza), on the day of preparation and dissociation (DIV 0). 4 million dissociated cortical cells were electroporated with 3 μg total DNA per condition using the program DC-104, seeded in six-well plates in DMEM/GlutaMAX supplemented with 5% FBS and incubated for 4 h and then replaced with neuron culture media and incubated at 37°C until harvesting. The following amounts of DNA were used for the relevant nucleofections: 2 μg chimeric miR30a-miR329 and 495 hairpins for miRNA overexpression validation; 2 μg pSUPER and 1 μg GFP for protein quantifications upon Prr7 knockdown; 2 μg HA-Prr7 and 1 μg GFP for validation of Prr7 overexpression.

HEK293T cells were transfected in HEK media supplemented with Hepes (25 mM), at 1.9 million cells seeded in 6-cm dishes per condition, by combining DNA with polyethylenimine (PEI) and Opti-MEM for 15 min and then adding the mixture dropwise onto cells. Cells were incubated at 37°C for 2 d until harvesting. For HA-Prr7 validation, cells were transfected with 2 μg pcDNA, HA-Cav1.2, or HA-Prr7, and 1 μg GFP. For shRNA-resistant mutant validation, cells were transfected with 100 ng HA-Prr7 constructs, 1 μg pSUPER, and 1 μg GFP.

## Stimulation

To examine downscaling processes, DIV18 or DIV19 hippocampal cells were stimulated with either picrotoxin (100 μM; Sigma-Aldrich) or equivalent volume (1:500) of ethanol absolute for 48 h.

For investigating CDK5 inhibition, DIV19 hippocampal cells were stimulated with either roscovitine (10 μM, R7772; Sigma-Aldrich) or equivalent volume (1:1,000) of DMSO for 18 h.

For studying SPAR and GluA1 degradation, DIV5 cortical cells were stimulated with either leupeptin (200 μg/ml, L2884; Sigma-Aldrich) or equivalent volume (1:250,000) of water for 20–21 h.

## Luciferase reporter assay

DIV13 primary rat hippocampal neurons were transfected in triplicate with 20 pmol pLNAs (10 pmol for 329/495 mix) and 50 ng Prr7 3′ UTR pmiRGLO constructs per well. Cells were treated with PTX or ethanol on DIV18 or DIV19 for 48 h, then lysed in Passive Lysis Buffer (diluted to 1×; Promega) for 15 min, and dual-luciferase assay performed using homemade reagents (as described in Baker and Boyce [2014]) on the GloMax Discover GM3000 (Promega).

pLNAs used were control pLNA (miRCURY LNA miRNA Power Inhibitor Negative control A, #339135 YI00199006-DCA; QIAGEN), miR-329 pLNA (miRCURY LNA miRNA Power Inhibitor RNO-MIR-329-3P, # 339130 YI04101481-DCA; QIAGEN), and miR-495 pLNA (miRCURY LNA miRNA Power Inhibitor HSA-MIR-495-3P, #339130 YI04101229-DCA; QIAGEN).

## Bicistronic reporter (dual-color) assay/single-cell fluorescent sensor assay

For single-cell fluorescent assay, DIV13 or DIV14 hippocampal cells were transfected in triplicate wells with 125 ng of control, miR-329,

or miR495 bicistronic sensor and 5 pmol control, miR-329, or miR-495 pLNA (where applicable). On DIV19, cells were treated with PTX or ethanol for 48 h, then fixed for 15 min in 4% paraformaldehyde/4% sucrose/PBS, washed in PBS, and directly mounted onto slides for imaging. To determine miRNA activity, dsRED-positive (red or yellow) versus GFP-only (green) cells were manually counted at 20× objective with both 488 and 561 channels open for all coverslips, and the proportion of GFP-only cells over the total count was taken. Approximately 100–200 cells were counted per coverslip (300–600 total per experimental condition).

### Immunocytochemistry, spine density, and image analysis

For all imaging experiments, hippocampal cells were transfected on DIV13 in either duplicate or triplicate wells. The following amounts of DNA/RNA were used for the relevant experiments: 150 ng GFP-amp, 20 pmol pLNAs (10 pmol for 329/495 mix), 500 ng miR30a hairpins, 7.5 ng pSUPER constructs (with the exception of spine rescue experiment with pLNAs, for which 2.5 ng was used), 400 ng HA-Prr7 or HA-Prr7$^R$, 400 ng CREB-VP16m (in active CREB variant for control purposes), or CDK5-DN. Where applicable, the transfected cells were further treated on DIV19 with picrotoxin/ethanol or roscovitine/DMSO.

For all experiments, cells were fixed for 15 min with 4% paraformaldehyde/4% sucrose/PBS and washed with PBS. In cases where cells were only analyzed for spine morphology, coverslips were directly mounted onto microscope slides.

For immunostaining, following fixation, coverslips were transferred to a humidified chamber protected from light. For Prr7 and whole-cell GluA1 immunostaining, cells were permeabilized with 0.1% Triton/PBS for 5 min. Blocking for 30 min in 1×GDB buffer (0.02% gelatin/0.5% Triton X-100/PBS) was followed by overnight incubation with primary antibody in GDB at 4°C. Secondary antibodies in GDB were applied for 45 min. Coverslips were washed with PBS before and after fixation and application of each antibody and briefly in MilliQ before slide mounting. For GluA1 surface staining, cells were treated with primary antibody at 37°C for 3 h. After washing the cells four times with fresh cell media, cells were fixed for 15 min with 4% paraformaldehyde/4% sucrose/PBS and washed with PBS. Coverslips were then transferred to a humidified chamber at room temperature, incubated in secondary antibody in GDB for 1 h, washed with PBS, rinsed briefly with MilliQ water, and mounted onto glass slides for imaging. For PSD-95/Synapsin1 co-staining, 0.1% Triton/10% NGS/PBS solution was added to the coverslips for 15 min and then incubated overnight with primary antibodies diluted in the Triton/NGS/PBS solution at 4°C. Secondary antibodies in Triton/NGS/PBS were applied for 1 h. The following primary antibodies were used: rabbit polyclonal anti-Prr7 (200 ng/ml, PA5-61266; Invitrogen), chicken monoclonal anti-Map2 (1:5,000 dilution, PA1-16751; Thermo Fisher Scientific), rabbit polyclonal anti-SPAR (1:1,500 dilution, kind gift of DT Pak), rabbit polyclonal anti-GluA1 (PC246 Calbiochem EMD Biosciences, or ABN241 Sigma-Aldrich at final concentrations 1 $\mu$g/ml for whole-cell staining, 2 $\mu$g/ml for surface staining), rabbit anti-Synapsin1 (AB1543, 1:1,000; Merck Millipore), and mouse anti-PSD-95 (810401, 1:200; BioLegend). Alexa-546– and -647–conjugated secondary antibodies (1:2,000 dilution) were used for detection.

All images were acquired with confocal laser-scanning microscope (Zeiss LSM) using a 40×/1.3 oil DIC UV-IR M27 objective. Z-stack images were obtained for 7–11 GFP-positive neurons with pyramidal morphology for each condition. Settings were Opt sampling (1.0× Nyquist), Zoom factor 1.0, Pixel Dwell 0.90 $\mu$s, Speed fps 0.23, Scan time 4.32 s, Speed 6, Digital Gain 1.0, and Pinhole 384 $\mu$m. Z stacks were kept at 0.45 $\mu$m, and 9–11 slices obtained. Laser settings were kept constant between conditions. Images were processed by Airyscan processing at 6.0 strength 3D, and maximum intensity projections of the Z-stacks were used for signal quantification.

Prr7, GluA1, and SPAR puncta intensities, PSD-95/Synapsin co-cluster number and surface GluA1 particle number within cell area were analyzed with a custom-made Python script developed by D Colameo and can be added as a Plugin on Fiji (https://github.com/dcolam/Cluster-Analysis-Plugin). Whole cell, cell body, and dendrites (whole-cell selection with cell body subtracted) were defined using GFP as a mask.

Spine density was measured manually in a blinded manner using Fiji (imaging was not blinded; however, cells were selected based on their pyramidal morphology at 10× objective where spines were not clearly visible). For each cell analyzed, first a primary dendrite and two secondary dendrites branching off it were selected, and the total length of the selected segment obtained. The selected segment was followed to the very end/edge of the frame, yielding a total segment length of ~150–250 $\mu$m, and therefore, the analysis was performed on both proximal and distal dendrites. Using the "cell counter" tool, the total protrusion number along the selected segment was counted (without discrimination of shape and included filopodia-like protrusions), yielding total counts of 100–300 protrusions. The count was divided by the total selected dendrite length to obtain #spines/$\mu$m. A second set of dendritic segments were selected, and the process repeated. The average of the two spine density readings was calculated per cell.

### Preparation of protein extracts and Western blotting

Protein extracts were prepared by first scraping and lysing cells in RIPA buffer (150 mM NaCl, 1% Triton X-100, 0.5% sodium deoxycholate, 1 mM EDTA, 1 mM EGTA, 0.05% SDS, 50 mM Tris, pH 8.0, 1× complete protease inhibitor cocktail [Roche]), spinning down the lysate at maximum speed at 4°C for 15 min, and collecting the supernatant. Protein concentration was measured using the Pierce BCA Protein Assay Kit (Thermo Fisher Scientific). Equal amounts of protein were diluted in Laemmli sample buffer supplemented with BME, boiled at 95°C for 5, min and loaded onto SDS–PAGE gels (10% polyacrylamide for the Prr7 probe, 8% for SPAR). For Prr7 Western, proteins were transferred onto Trans-Blot Turbo 0.2-$\mu$m nitrocellulose membranes (Bio-Rad) using the Trans-Blot Turbo semi-dry transfer system (Bio-Rad). For SPAR Western, proteins were blotted onto 0.45-$\mu$m PVDF membranes (Immobilon) soaked in transfer buffer (25 mM Tris–HCl, pH 8.3, 192 mM glycine, 20% MeOH) via wet transfer for 15–16 h at 25V. For all experiments, blocking was done in 5% milk in 1xTBS 0.1% Tween20 (TBST) for 1.5–2 h at room temperature, followed by overnight primary antibody incubation at 4°C. After washes in milk, HRP-conjugated secondary antibodies were applied onto the membranes for 1 h at room temperature. Membranes were washed in TBST and visualized with the Clarity Western

ECL Substrate (Bio-Rad) on the ChemiDoc Imaging System (Bio-Rad). The following primary antibodies were used: rabbit anti-GluA1 (1:1,000 dilution, PA1 37776; Thermo Fisher Scientific), mouse monoclonal anti-Prr7 (1:250 dilution, MA1-10448; Thermo Fisher Scientific), and rabbit monoclonal anti-$\alpha$ tubulin (1:2,000 dilution, 11H10 lot 112125S; Cell Signaling). HRP-conjugated secondary antibodies rabbit anti-Ms IgG H&L (402335 lot D00160409; Calbiochem) and goat anti-Rb IgG H&L (lot 2625715; Calbiochem) were used at 1:20,000 dilution.

### RNA extraction and quantitative real-time PCR

RNA was isolated using the TriFast RNA extraction kit (30-2030; VWR) or RNA-Solv reagent (Omega Bio-tek). Genomic DNA was removed with TURBO DNAse enzyme (Thermo Fisher Scientific). Reverse transcription was performed using either the Taqman MicroRNA Reverse Transcription Kit (Thermo Fisher Scientific) for miRNA detection or the iScript cDNA synthesis kit (Bio-Rad) for mRNA detection. qPCR was performed using either Taqman Universal PCR Master Mix (Thermo Fisher Scientific) for microRNA detection or the iTaq SYBR Green Supermix with ROX (Bio-Rad), and plates were read on the CFX384 Real-Time System (Bio-Rad). Data were analyzed via the $\Delta\Delta$Ct method and normalized to either U6 (for miRNAs) or GAPDH (for mRNAs). mRNA primer information is indicated in supplemental table.

Taqman primers used were (all from Thermo Fisher Scientific): U6 snRNA (Assay ID: 001973), mmu-miR-495 (4427975, Assay ID: 001663), mmu-miR-329 (4427975, Assay ID: 00192), mmu-miR-134 (4427975, Assay ID: 001186), hsa-miR-132 (4427975, Assay ID: 000457), and hsa-miR-99b-5p (4427975, Assay ID: 000436).

### Electrophysiology

Whole-cell patch-clamp recordings were performed on an upright microscope (Olympus BX51WI) at room temperature. Data were collected with an Axon MultiClamp 700B amplifier and a Digidata 1550B digitizer and analyzed with pClamp 11 software (all from Molecular Devices). Recording pipettes were pulled from borosilicate capillary glass (GC150F-10; Harvard Apparatus) with a DMZ-Universal-Electrode-Puller (Zeitz) and had resistances between 3 and 4 MW.

Miniature EPSC (mEPSC) were recorded from primary cultured hippocampal neurons on DIV19-21 after transfection in Neurobasal-A medium (10888022; Thermo Fisher Scientific) on DIV9-12. The extracellular solution (ACSF) was composed of (in mM) 140 NaCl, 2.5 KCl, 10 Hepes, 2 CaCl$_2$, 1 MgCl$_2$, 10 glucose (adjusted to pH 7.3 with NaOH), the intracellular solution of (in mM) 125 K-gluconate, 20 KCL, 0.5 EGTA, 10 Hepes, 4 Mg-ATP, 0.3 GTP, and 10 Na$_2$-phosphocreatine (adjusted to pH 7.3 with KOH). For mEPSCs, 1 $\mu$M TTX and 1 $\mu$M Gabazine were added to the extracellular solution to block action-potential driven glutamate release and GABAergic synaptic transmission, respectively. Cells were held at −70 mV. The sampling frequency was 5 kHz, and the filter frequency 2 kHz. Series resistance was monitored, and recordings were discarded if the series resistance changed significantly (≥10%) or exceeded 20 MΩ. Paired pulse ratio of EPSCs were recorded at a holding potential of −60 mV, sampled at 100 kHz, and filtered at 4 kHz. 1 $\mu$M Gabazine was added

to the extracellular solution to block GABAergic synaptic transmission. 100 nM NBQX was added to prevent epileptiform activity and to minimize polysynaptic activity. Synaptic currents were evoked by monopolar stimulation with a patch pipette filled with ACSF.

### Statistics

With the exception of the miRNA sensor assay, statistical tests were performed using GraphPad Prism version 9.2.0. For all datasets, three to four independent experiments were performed. Given the small sample size, normality was assumed for all datasets, and therefore, one or two sample unpaired $t$ test (two-sided) or one-way or two-way ANOVA followed by a post hoc Tukey test was performed. *$P < 0.05$; **$P < 0.01$; ***$P < 0.001$.

For the miRNA sensor assay, a binomial generalized mixed effects model (GLMM) was applied (per sensor) using the lme4 R package (Bates et al, 2015) to test the proportion data (ratio of miRNA positive cell counts over all cells). Total counts were used as prior weights. The preparation (e.g., the batch) was accounted for using a random effect. Because of the replicates within the batches, an interaction term between the (random) batches and the (fixed) treatments was included in the model. Post hoc tests were conducted using the glht function of the multcomp package (Hothorn et al, 2008) for the comparisons of interest. $P$-values were adjusted by Bonferroni's method.

## Supplementary Information

## Acknowledgements

We would like to thank DT Pak and M Sheng for generously providing the SPAR antibody and CDK5-dominant negative vector and ME Greenberg for the CREB-VP16m construct. We thank M Soutschek, R Fiore, S Bicker, and C Gilardi for cloning the pmiRGLO, miR-329/control sensors, control pSUPER, and miR30a-Ctr hairpin constructs, respectively. We are grateful for R Fiore for extensive discussions regarding the manuscript and E Sonder for help with statistical analysis. We thank the excellent technical assistance of T Wüst and C Furler. This work was supported by a grant from the Swiss National Foundation (SNF 310030_205064) to G Schratt.

### Author Contributions

MO Inouye: investigation, methodology, and writing—original draft, review, and editing.
D Colameo: software and methodology.
I Ammann: investigation.
J Winterer: investigation.
G Schratt: conceptualization, supervision, funding acquisition, project administration, and writing—original draft, review, and editing.

**Conflict of Interest Statement**

The authors declare that they have no conflict of interest.

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
