## [Reviewer comments · Life Science Alliance]

Life Science Alliance

miR-329 and miR-495-mediated Prr7 downregulation is required for homeostatic synaptic depression

Michiko Inouye, David Colameo, Irina Ammann, Gerhard Schratt, and Jochen Winterer

DOI: <https://doi.org/10.26508/lsa.202201520>

Corresponding author(s): *Gerhard Schratt, ETH Zurich*

Review Timeline:	Submission Date:	2022-05-10
	Editorial Decision:	2022-05-12
	Revision Received:	2022-08-11
	Editorial Decision:	2022-09-09
	Revision Received:	2022-09-13
	Accepted:	2022-09-13

Transaction Report:

Please note that the manuscript was reviewed at Review Commons and these reports were taken into account in the decision-making process at Life Science Alliance.

May 12, 2022

Re: Life Science Alliance manuscript #LSA-2022-01520

Prof. Gerhard Schratt
University of Marburg
Physiological Chemistry
Karl-von-Frisch Str. 1
Marburg 35037
Germany

Dear Dr. Schratt,

Thank you for submitting your manuscript entitled "miR-329 and miR-495-mediated Prr7 downregulation is required for homeostatic synaptic depression in rat hippocampal neurons" to Life Science Alliance. We invite you to re-submit the manuscript, revised according to your Revision Plan.

Thank you for this interesting contribution to Life Science Alliance. We are looking forward to receiving your revised manuscript.

Sincerely,

B. MANUSCRIPT ORGANIZATION AND FORMATTING:

Inouye et al., LSA, Point-by point response letter

Reviewer #1 (Evidence, reproducibility and clarity (Required)):

Inouye et al. study the role of Prr7 in homeostatic synaptic depression (HSD) in rat neurons using an established model based on piritoxin (PTX) treatment. Prr7 mRNA and protein levels decrease in the synaptic-dendritic compartment upon PTX treatment, which is essential for PTX-mediated decrease in spine number and GLUA1 levels. The authors furthermore demonstrate that Prr7 levels are under the control of miR-495 and miR-329 that are also regulated via PTX. One way by which PRR7 affects HSD appears to be via the regulation of SPAR and more precisely by preventing SPAR from phosphorylation by CDK5. Overall, this is a very interesting manuscript and the presented data are easy to follow and of a refreshing clarity. The only issue may be the last piece of data related to the role of Cdk5 which is not yet at the same level as the analysis of Prr7 the corresponding upstream mechanisms.

Here are some questions that came up when reading the manuscript.

- Fig 1G,H. The quality of the images could be improved. Map2 staining is not really visible and does not seem to overlap with GFP or PRR7 signal. Also, the decrease in whole cell vs. dendrites seems to be identical and I wonder if the same bar has been pasted by mistake. Moreover, from the representative image the reduction of PRR7 in the cell body appears as pronounced as in the dendrites.

> We have improved image quality by increasing the MAP2 signal and decreasing the GFP signal. However, we want to note that Prr7 is also reduced in the cell body compartment, although to a slightly lesser extent (Fig. 1C-G), which is perfectly mirrored in the Prr7 panels of the representative pictures.

- Fig. 3D. The fact that mir329 is not significant in the luciferase assay may be due to the slightly higher variability in the data, when compared to the findings on miR-495, that in fact look very similar. I suggest to increase the sample size for both experiments. This may also help to clarify if there is indeed an additive effect, which is based on the data shown in panels B-D a bit difficult to appreciate, although the data shown in panel E seems to support this view.

> based on the expected effect size and variability of the data, a power analysis suggested N=4 as the appropriate sample number for these experiments. We also want to note that the experiments from two independent readouts (luciferase reporter assays, Fig. 3B-D; immunostaining, 3E-F) perfectly align, namely showing a slightly higher effect for miR-495 compared to miR-329 inhibition alone, and a complete reversal of the PTX effect by a combined miR-329/495 inhibition. Therefore, we did not see the necessity to perform additional luciferase assays.

- Is the data from the miR-sensor experiment shown in Fig. 3H significant? If not I would again increase the "n" to see if this is simply due the small samples size.

> we have now performed a statistical assessment of the data provided in original Fig. 3H (new Fig. 3G). This analysis shows a highly significant increase in miR329/495 activity upon PTX treatment, which is lost if the corresponding anti-miRs (329/495 pLNAs) were co-transfected.

- Unlike the other experiments on the miRs and Prr7, the data related to the function on CDK5 is solely based on a pharmacological inhibitor that is, however, not specific to Cdk5.

> we agree that roscovitine is a broad-spectrum inhibitor of several Cdks and have therefore adjusted the interpretation of this data in the text. Furthermore, we have introduced a second means to interfere with CDK5 activity more directly using a previously published dominant-negative construct (CDK5D144N, Seeburg et al., Neuron 2008; new Fig. 5D). Thereby, we observed that CDK5DN144N interfered with both the PTX-dependent reductions in spine density and dendritic SPAR signal intensity in a very similar way as roscovitine. This provides further evidence for an important role of CDK5 in Prr7-mediated HSD.

- The effect of Cdk5 may need some more attention, since inhibition of knock down of CDK5 activity/levels is known to decrease spine density. This might also be observed in Fig. 5F when comparing the Ctr groups. In sum, the data on Cdk5 is not as convincing as the other data and also the fact that reduced SPAR protein levels was no longer seen in Prr7 shRNA-transfected cells treated with Roscovitine is likely due to one outlier amongst the 3 independent samples analyzed.

> we disagree with the interpretation of the data from original figures Fig. 5E and 5F (new Fig. 5B, C) provided by this reviewer. Although we observe a slight (and expected) reduction of spine density with roscovitine (new Fig. 5C, left graph) in the control shRNA condition, this is not significant. Therefore, the main effect of roscovitine is clearly to prevent spine size reduction in the Prr7 shRNA condition. The potential "outlier" in new Fig. 5C (right graph, Prr7 shRNA + roscovitine condition) is actually in the opposite direction, meaning that removing it would shift the data even to higher SPAR levels. Overall, there is a clear lack of SPAR downregulation by Prr7shRNA in the context of roscovitine, even considering the variability of the data.

Reviewer #1 (Significance (Required)):

The manuscripts describes a novel pathway in HSD, which will be highly interesting to the neuroscientific community and the field of microRNA biology in general. Thus, the data is very significant

Reviewer #2 (Evidence, reproducibility and clarity (Required)):

The article "miR-329 and miR-495-mediated Prr7 downregulation is required for homeostatic synaptic depression in rat hippocampal neurons" by Inouye et. al. highlighted the importance of miRNA-mediated regulation of dendritic spine density at the post-transcriptional level during synaptic downscaling. The manuscript provided a biochemical evidence for miRNA-driven localized changes in dendritic spines that plays a pivotal role in adjusting the synaptic strength upon chronic hyperactivity. The study showed specific enhancement of miR-495 level in neuronal dendrites upon chronic hyperactivity. Authors have demonstrated that the prolong treatment of PTX regulates the activity of miR-329 and miR-495 in neuronal dendrite. This study demonstrated a direct correlation between the enhanced miRNA activity and reduction of dendritic spines during homeostatic synaptic depression (HSD). The study provided a mechanistic insight of HSD - induced reduction of dendritic spine via CDK5-SPAR pathway involving reduction of spine-associated GTPase SPAR possibly through proteasomal degradation. Although, the study identified a localized biochemical mechanism influencing dendritic spine structure upon chronic hyperactivity, the significance of this regulatory in HSD should be bolstered by electrophysiology experiments. Overall, miRNA-mediated localized regulation of synaptic scaling is potentially interesting but current manuscript lacks important experimental support for the claim. I must agree that authors presented all data set with clarity and research methodology explained adequately. My comments for further improvement of the manuscript are as follows:

1) The authors should use whole-cell patch clamp recording and measure mEPSC amplitude and frequency upon knockdown of Prr7. This should mimic PTX-induced reduction of mEPSC amplitude reported by Schrott group and others. To further pinpoint the role Prr7 in HSD, mEPSCs should be measured from hippocampal neurons overexpressing Prr7 following PTX treatment. Ideally, overexpression of Prr7 should rescue (at least partial) reduction in mEPSC amplitude. Similarly, mEPSCs should be measured from PTX-treated hippocampal neurons transfected with inhibitors of miR-495 and miR-329.

>as suggested by this reviewer, we have now performed patch-clamp recordings of mEPSC in our rat hippocampal neuron model. We started these experiments with Prr7 shRNA transfected neurons, since we expected the most robust effect on mEPSC frequency given our previous results from spine density measurements (Fig. 2A, B). Surprisingly, we did not observe significant alterations in both mEPSC frequency and amplitude in Prr7 shRNA compared to control shRNA-transfected neurons (new suppl. Fig. 7a), suggesting that observed morphological effects do not translate into corresponding electrophysiological changes at synapses. Consistent with the ephys data, we also did not find differences in the density of excitatory synaptic co-clusters (PSD-95/synapsin) between Prr7 and control shRNA transfected neurons (suppl. Fig. 7c) One possible explanation, in particular considering previous reports about Prr7 secretion, could be non-cell autonomous presynaptic effects masking altered

postsynaptic function, e.g. reduced spine numbers. Such a view is supported by our results demonstrating altered paired-pulse ratio (PPR), a parameter which is typically affected by changes in presynaptic function (suppl. Fig. 7b). Alternatively, and not mutually exclusive, Prr7 knockdown might lead to a loss of spine-associated excitatory synapses, which is compensated for by an increased number of excitatory synapses forming onto the dendritic shaft and/or cell body. We have provided a discussion of these possibilities in the discussion section of the revised manuscript.

2) Is Prr7 transported to dendrite from cell body upon PTX treatment? This could be additional regulatory control apart from miRNA-mediated regulation of Prr7.

> We agree with this reviewer that our experimental setup monitoring endogenous Prr7 mRNA changes does not allow us to unequivocally distinguish PTX effects on mRNA stability from those on mRNA transport. Based on our previously published RNA-seq data from compartmentalized neuron cultures (Colameo et al., EMBO Rep. 2021), Prr7 mRNA is enriched in dendrites under basal conditions, suggesting an active dendritic transport mechanism for Prr7 mRNA. However, we observe a robust downregulation of Prr7 mRNA in dendrites and, albeit to a lesser extent, also in cell bodies, upon PTX, making it unlikely that PTX-mediated Prr7 mRNA changes are solely due to a re-distribution of Prr7 mRNA by active transport (in which case one would expect that PTX changes Prr7 mRNA in the different compartments in opposite directions).

Similarly, authors need to evaluate localized maturation of miR-495 upon PTX treatment. This can be tested in Dicer knockdown background.

> We agree with this reviewer that a potential localized maturation of miR-495 would represent an attractive mechanism to explain the dendritic upregulation of miR-495 expression upon PTX. However, assessing local miRNA processing is not trivial and would require establishing fluorescent pre-miR sensors in combination with local stimulation (Sambandan et al., Science 2017), which would represent a PhD project on its own and is therefore clearly beyond the scope of this revision.

3) Authors emphasized that Prr7 expression is post-transcriptionally regulated during HSD. This claim should be supported by data showing the Prr7 expression at the transcript level is not altered by inhibition of transcription during PTX treatment.

> We agree that based on our current data, we cannot rule out that reduced transcription of Prr7 upon PTX contributes to the observed Prr7 mRNA downregulation. The experiment suggested by the reviewer however is in our opinion not suitable to address this point for two reasons: first, blocking transcription would already lead to reduced Prr7 levels under basal conditions, which would make an interpretation of the PTX effect difficult. Second, using transcriptional inhibitors (e.g. actinomycin) in neurons for

extended periods (e.g. during 48h PTX) tends to be very toxic in our experience and can lead to massive cell death, which also would severely confound the results.

Therefore, we instead looked at intronic reads of the Prr7 pre-mRNA in the cell body compartment of our previously published RNA-seq dataset (Colameo et al., 2021).

Changes in pre-mRNA reads can serve as a proxy for potential transcriptional effects.

However, Prr7 pre-mRNA reads were not different in the cell body compartment between PTX- and mock-treated neurons, strongly arguing against an important contribution of transcriptional inhibition to the observed reductions in Prr7 mRNA levels.

4) Authors anticipate that the Prr7 protects SPAR protein from Plk2-mediated degradation. If this anticipation is correct, PTX treatment in presence of proteasome inhibitor should protect SPAR.

> it is already known that SPAR is degraded in a proteasome-dependent manner (Pak and Sheng, Science 2003; Lee et al., Neuron 2011). We therefore decided to focus on the question whether proteasome-dependent degradation of SPAR is indeed downstream of Prr7. To address this, we investigated if proteasome inhibition prevents the decrease of Prr7 and downstream targets (e.g. GluA1) induced by Prr7 knockdown. Unfortunately, the use of a standard proteasome inhibitor (MG-132) in primary neurons for 48h was highly toxic even at very low concentrations (10 nM), making the results from these experiments non-interpretable. As an alternative, we used instead leupeptin, a more broad-spectrum protease inhibitor which primarily inhibits serine and cysteine proteases (e.g. lysosomal proteases), but also displays activity against the trypsin-like activity of the proteasome (Harer et al., The Journal of Antibiotics, 2012). 48h leupeptin treatment was well tolerated by primary rat hippocampal neurons and led to a nearly complete reversal of the SPAR and GluA1 degradation induced by Prr7 knockdown. These results demonstrate that the effects of Prr7 shRNA on SPAR/GluA1 are dependent on protein degradation. Candidate proteases are either the proteasome, which is involved in PTX-mediated SPAR degradation (see above), or the lysosome, which was shown to participate in the activity-dependent degradation of GluA1-containing AMPARs (Schwarz et al., J. Neurosci 2010). We have now discussed these possibilities in the revised manuscript.

Figure 5B photomicrograph should indicate molecular weight marker. The merge of molecular weight marker and SPAR blot may not be acceptable to journal of choice.

>we have now changed Figure 5B according to the suggestion of the reviewer

5) It is not clear to me why authors did not test the level of surface GluA2 following Prr7 knockdown. The experimental evidence will show specificity of GluA1 or GluA2 or requirement for both. GluA1 and GluA2 surface expression has been shown to be affected by chronic hyperactivity as reported by many groups including publication from Schrott laboratory.

>the decision to focus on GluA1 rather than GluA2 was based on preliminary results from Western blots where we observed a strong downregulation of GluA1, but not GluA2 (Fig. 1 below), upon Prr7 knockdown. This result suggests that Prr7 works in a branch of the HSD signaling pathway which specifically targets the GluA1 subunit for internalization. Given the lack of effect on total GluA2 levels by Prr7 knockdown, we felt that experiments addressing the surface expression of GluA2 were not of the highest priority.

Fig. 1: Western blot with lysates from primary rat cortical neurons nucleofected with the indicated shRNAs and probed with antibodies against beta-tubulin (upper panel; loading control) and GluA2 (lower panel).

6) Sample size is too low for all surface GluA1 and spine density analysis. How many neurons were analyzed? Authors mentioned "n=3" in Figure 2 (spine and sGluA1 analysis). It may be 3 independent experiments and does not reflect the numbers of neurons. If so, please mention the number of neurons analyzed. Most figures shows only 3 independent experiments used for statistical significance. Authors should consider adding more data point. It could be possible that addition of more data point for figure 2D may show statistical significance for luciferase expression following inhibition of miR-329.

>we apologize for the confusion about sample sizes and have now provided a more detailed description in the respective figure legends. Concerning the experiments on sGluA1 and spine density analysis, in each of the three independent experiments, at least 10 neurons from 2 replicate coverslips were analyzed. Comparable sample sizes have been used by us and others in the past (e.g. Fiore et al., EMBO J. 2014; Rajman et al., EMBO J. 2017) and where sufficient to obtain robust results (based on the observed effect size and variance of the data). We therefore don't see the necessity to increase sample size for spine density and sGluA experiments. Regarding luciferase data (Fig. 3 B-D), please refer to our comment for reviewer 1.

7) Exosomal secretion of Prr7 reduces spine size. Does Prr7 secreted out from neuronal dendrite upon PTX treatment? It could be a possibility apart from miRNA-mediated downregulation. Authors should discuss this in discussion section.

> a putative Prr7 secretion from neuronal dendrites is indeed an exciting possibility based on a previous publication (Lee et al., Nature Comm. 2018). We have devoted a separate paragraph on the potential implications of secreted Prr7 for the observed morphological and electrophysiological changes in the discussion of the revised manuscript.

8) Quantitative analysis for Figure 5G is important

>the data of former Fig. 5G (new Fig. 5C) was already quantified in the original version of the manuscript (Fig. 5E, F), but this was probably overlooked by the reviewer due to the order of the panels. We apologize for this and now show the representative images before the respective quantification in the new Fig. 5C of the revised manuscript.

9) Statistical significance between respective conditions should be indicated in all figures. It is included in many figures but some lacks this info. Example: Fig 3H

>according to this comment, we have now provided a statistical assessment of the data presented in former Fig. 3H (new Fig. 3G) in the revised manuscript.

10) Photomicrograph and plot in Figure 1C and 1D is not matching. Please include correct photomicrograph for Prr7.

> we want to stress that the quantifications provided in Fig. 1C and D of the revised manuscript represent the average of 3-4 independent experiments. Nevertheless, we tried our best to choose a representative blot which matches the quantifications. All other blots can be found in the source data files.

11) GluA1/A2 internalization by SPAR should be tested to demonstrate the connection between Prr7 downregulation by miRNAs and surface expression of GluA1/A2.

- **We felt that such an experiment was not of high priority, since the connection between SPAR and GluA1/2 was already established previously (Lee et al., Neuron 2011)**

12) It is not clear to me why authors used one-sample t-test in Figure 1 instead of unpaired student t-test?

> the data in Fig. 1 is presented as a ratio between PTX and mock-treated neurons. To assess whether any of the conditions significantly differs from the baseline (1), a one-sample t-test must be used.

Minor comments:

1) Authors mentioned in discussion (Page 12) that Prr7 could be "newly" synthesized in neuronal dendrites and could be measured by puro-PLA technique. I am bit confused here. Prr7

is downregulated by miRNAs. Therefore, one can use Puro-PLA to show reduced synthesis following PTX treatment. This discrepancy should be corrected.

> we thank the reviewer for this comment and have now indicated in the discussion that Puro-PLA could be used to demonstrate a local decrease in Prr7 synthesis upon PTX.

2) Authors mentioned "interference of miR-329 RISC binding to the Prr7 3'UTR by an RNA-binding protein which is removed upon PTX treatment". This is an interesting possibility as shown by Srinivasan et. al. PLoS Biology 2021 and other references cited in the manuscript. How authors envisage miR-329 RISC binding to Prr7 3'UTR is influenced by PTX treatment? Does miR-495 RISC binding differ with miR-329 RISC binding upon PTX treatment? These points should be discussed with more clarity in the discussion section.

> following the suggestion of this reviewer, we have now discussed in more detail the potential mechanism of RBP-miR-329 crosstalk at the Prr7 3'UTR, focusing on a putative competition between miR-329 and CELF for a common motif in the Prr7 mRNA.

Reviewer #2 (Significance (Required)):

Significance:

The manuscript proposed a local biochemical regulation of miRNA-induced synaptic downscaling. Although, the concept exists in the field (Sutton and Schuman Cell 2006 and Sutton and Schuman Neuron 2007), this study provides a non-coding RNA-mediated local control of homeostatic structural plasticity. In my opinion, this study supports the previous concept but not a significant conceptual advance. Also, authors need to use electrophysiology to support their claim. The electrophysiology data would strengthen the manuscript that will be appreciated by the field. If authors can address major concerns, the manuscript would attract attention in the homeostatic plasticity field both at the molecular and systems level. This study has the potential to explain some of the system level findings that show local dendritic function necessary for tuning the synaptic strength (Letellier and Goda PLoS Biology 2019). This study has potential to open up new directions to study how localized protein synthesis and degradation by miRISC could influence synaptic scaling (Srinivasan et. al. PLoS Biology 2021).

My expertise is in the field of RNA based mechanisms of homeostatic and Hebbian forms of synaptic plasticity. We employ whole cell patch clamp recording, confocal microscopy, subcellular fractionation and biochemical analysis to understand plasticity mechanisms operating at the subcellular neuronal compartments.

****Referees cross-commenting****

There are overlapping comments among all three reviewers, in particular Fig 3 and 5 and sample size. Also, patch clamp recording is important to pinpoint miR495/Prr7 in scaling is necessary.

Reviewer #3 (Evidence, reproducibility and clarity (Required)):

Summary:

miR-329 and miR-495-mediated Prr7 downregulation is required for homeostatic synaptic depression in rat hippocampal neurons by Inouye et al., explores molecular mechanisms of synaptic scaling. Authors discover new connection between RNAs and proteins that have been previously implicated in the process.

Overall the pathway described is novel and takes into account connections between the following elements:

miR-329/495 ==| Prr7 ==| CDK5 ==| SPAR ==> dendritic spine elimination during HSD induced by chronic activity

The Schrott group is an excellent group in molecular neuroscience with great standing. The study is scientifically and technically sound.

Major comments:

(1) The work depends on a candidate gene-by-gene approach. For this reason, the overall involvement of the reported pathway in HSD and the roles of individual biomolecules in the signaling and regulatory cascade do not reveal if these are indeed key players (or secondary adjuvant factors) in local posttranscriptional control of HSD in neurons.

>we are confident that a candidate gene approach as the one presented here is able to identify important players and pathways in HSD. Such approaches are complementary to more unbiased screening approaches, which have been performed by us (Rajman et al., 2017; Colameo et al., 2021) and others (e.g. Schanzenbacher et al., Neuron 2016) in the past. In fact, some of the molecules studied here (miR-495, Prr7) have been independently identified by transcriptomics/proteomics screens (Rajman et al., 2017; Colameo et al., 2021).

(2) Modest to moderate biological effects (that are statistically significant) should encourage authors to provide orthogonal means to justify the biological centrality of some of the effects measured.

> we want to stress that the biological effects observed by us (e.g. spine density reduction of 20%) correspond to the ones routinely observed in the field of homeostatic synaptic scaling (e.g. Seeburg et al., Neuron 2008). In fact, reducing synaptic input to larger degrees might be potentially harmful to neurons and therefore not be useful in the context of synapse homeostasis. That being said, we still agree with this reviewer that providing multiple readouts for a given treatment (e.g. Prr7 knockdown) is important to rule out potential artefacts. For example, we already provided sGluA1 stainings (Fig. 2F) in the original manuscript in addition to spine density measurements. In the revised manuscript, we have now further included data from patch-clamp recordings of mEPSCs (Fig. S7a, b), PSD-95/Synapsin co-cluster analysis (Fig. S7c) and biochemistry (new Fig. 6).

(3) The work in mouse primary neurons with species specific miRNAs upstream of Prr7 suggests that if relevant to humans, other miRNAs might be involved. In more details: Targetscan suggests miR-329-3p/362-3p/miR-495-3p)but also miR-411-3p/ miR-377-3p(as regulators of Prr7 3'UTR. These miRNAs are not extensively conserved in evolution. In the human Prr7, two other highly conserved miRNAs seems to potentially be relevant : miR-455-3p.1 and miR-194-5p.

> we agree that miR-329-3p and miR-495-3p might not be the only miRNAs regulating Prr7 expression. However, based on our data, the combination of these two microRNAs is essential for PTX-dependent spine density and Prr7 reduction (Fig. 3E, 4A), suggesting that they play a particularly important role. The selection of these miRNAs was based on expression data from primary rat neurons (Khudayberdiev et al., 2013) and small RNA-seq of PTX-treated neurons (Rajman et al., 2017). Concerning the miRNAs potentially targeting the human Prr7, miR-455-3p has a very low expression level in rat neurons and therefore unlikely contributes significantly to Prr7 regulation. In contrast, miR-194-5p is highly expressed and its binding site fully overlaps with the miR-411/495 site based on TargetScan. This miRNA should therefore be considered in future studies which are however beyond the scope of this revision.

(4) The study relies on whole cellular approaches. Experiments to explore local regulation of protein synthesis in the dendritic spines may improve the sensitivity of testing some of the effects and provide more appropriate experimental means to the hypothesis that local compartmentalized regulation of protein synthesis is taking place.

Such approaches may address relevant questions such as: Where is Prr7 mRNA is actively translated? Where does miRNA silencing takes place?

> first, we would like to point out that several assays investigating compartmentalized effects have already been included in the original version of the manuscript. These include the use of compartmentalized neuron culture system (Fig. 1A-D, Fig. 3H, I) and the quantification of immunohistochemistry in cell bodies and dendrites (e.g. Fig. 1E, Fig. 5). Second, our attempts to establish Puro-PLA to more directly visualize local protein synthesis of endogenous Prr7 failed, probably due to the pure quality of the

commercially available Prr7 antibodies. Finally, transfection of GFP-Prr7-3UTR fusion constructs did not yield sufficient dendritic localization based on FISH, suggesting that critical sequence elements for dendritic transport of Prr7 were not included in these constructs. This precluded a further assessment of the role of miRNA silencing, e.g. by the use of corresponding miRNA binding site mutants. Accordingly, we have now toned down several statements regarding a local control of Prr7 synthesis by miR329/495 in neuronal dendrites.

(5) HSD assays: In several points in the manuscript, assays do not focus on homeostatic synaptic depression as a primary endpoint. In addition, orthogonal ways to evaluate HSD are encouraged (Surface GluR1 staining?; mEPSCs? Other means?).

> see our response to 2) of the same reviewer.

Reviewer #3 (Significance (Required)):

The work is novel and important and on a critical topic to molecular neuroscience

The methods are appropriate but biased and the analysis is very focused and can be substantiated by orthogonal approaches.

In addition, more unbiased pathway discovery approaches and robust assays might enable different and perhaps stronger effects to be discovered

****Referees cross-commenting****

My feeling is that the work is technically sound and novel.

I wish to have instructions and learn how to judge 'impact' and 'excitement' for review commons.

September 9, 2022

RE: Life Science Alliance Manuscript #LSA-2022-01520R

Prof. Gerhard Schratt
ETH Zurich
HEST
Winterthurerstr. 190
Zurich 8057
Switzerland

Dear Dr. Schratt,

Thank you for submitting your revised manuscript entitled "miR-329 and miR-495-mediated Prr7 downregulation is required for homeostatic synaptic depression". We would be happy to publish your paper in Life Science Alliance pending final revisions necessary to meet our formatting guidelines.

- please address Reviewer 2 and 3's remaining comments
- please add the Twitter handle of your host institute/organization as well as your own or/and one of the authors in our system
- please use the [10 author names, et al.] format in your references (i.e. limit the author names to the first 10)
- please add your supplementary figure legends to the main manuscript text, directly under the main figure legends
- please add panels a and b to your figure S4 figure legend
- please include approval details for the use of rats
- the primer sequences and qPCR primers listed in Supplemental Material would display better as Supplemental Tables

Figure Check:

- you may consider uploading Figure 7 as a Graphical Abstract instead
- please add scale bars to Figure S1B, S2B, S5C, S8, S12
- please add sizes next to all blots

A. FINAL FILES:

B. MANUSCRIPT ORGANIZATION AND FORMATTING:

Sincerely,

Reviewer #1 (Comments to the Authors (Required)):

Authors addressed all my concerns

Reviewer #2 (Comments to the Authors (Required)):

Authors have considered suggestion from reviewers seriously and address most of the concerns. The manuscript is now much improved and can be considered for publication after addressing very minor points mentioned below. The data presented in the manuscript strongly support their conclusions. I believe that the manuscript adds a new perspective and will be appreciated in the field.

Minor points:

1. Discussion on Prr7 transport needs to be included in the discussion section.

"Based on our previously published RNA-seq data from compartmentalized neuron cultures (Colameo et al., EMBO Rep. 2021), Prr7 mRNA is enriched in dendrites under basal conditions, suggesting an active dendritic transport mechanism for Prr7 mRNA. However, we observe a robust downregulation of Prr7 mRNA in dendrites and, albeit to a lesser extent, also in cell bodies, upon PTX, making it unlikely that PTX-mediated Prr7 mRNA changes are solely due to a redistribution of Prr7 mRNA by active transport."

2. Needs to be included in the discussion:

"Therefore, we instead looked at intronic reads of the Prr7 pre-mRNA in the cell body compartment of our previously published RNA-seq dataset (Colameo et al., 2021). Changes in pre-mRNA reads can serve as a proxy for potential transcriptional effects. However, Prr7 pre-mRNA reads were not different in the cell body compartment between PTX- and mock-treated neurons, strongly arguing against an important contribution of transcriptional inhibition to the observed reductions in Prr7 mRNA levels."

Reviewer #3 (Comments to the Authors (Required)):

This is a revised manuscript and the authors did an excellent job in addressing the previous questions. I strongly suggest to publish the study

I have only one minor comment the authors may want to address before publication. The authors should add to the discussion a few words on the numerous data implicating loss of physiological Cdk5 activity with impaired synaptic plasticity and synapse number and discuss this in the context of their data. The fact that in turn aberrant CDK5 activity can impair synapse function may also be interesting to consider in light of the presented data.

September 13, 2022

RE: Life Science Alliance Manuscript #LSA-2022-01520RR

Prof. Gerhard Schratt
ETH Zurich
HEST
Winterthurerstr. 190
Zurich 8057
Switzerland

Dear Dr. Schratt,

Thank you for submitting your Research Article entitled "miR-329 and miR-495-mediated Prr7 downregulation is required for homeostatic synaptic depression". It is a pleasure to let you know that your manuscript is now accepted for publication in Life Science Alliance. Congratulations on this interesting work.

DISTRIBUTION OF MATERIALS:

Again, congratulations on a very nice paper. I hope you found the review process to be constructive and are pleased with how the manuscript was handled editorially. We look forward to future exciting submissions from your lab.

Sincerely,
